# Structure of the *Enterobacter* pan-genome is revealed using machine learning

Joshua T. Burrows,[1] Gaoyuan Li,[1] Jonathan M. Monk,[1] Siddharth M. Chauhan,[1] Bernhard O. Palsson[1,2,3,4,5]

**ABSTRACT** The growing availability of publicly accessible *Enterobacter* genomes offers an opportunity to reveal the structure of its pangenome, uncovering the catalog of genes across the genus and their distribution across the different species and subspecies of the genus. In this study, we analyze 777 high-quality complete *Enterobacter* genomes using a pangenome matrix. The accessory genome, consisting of the genes found in many, but not all strains, was decomposed using non-negative matrix factorization (NMF) to identify groups of genes, called Phylons, that are found to be present across the subgroups of the genomes analyzed. The Phylons are representative of major modes of inheritance, both lineage-associated and horizontal, found across the pangenome. Using NMF, we defined 31 Phylons, representative of 21 lineage-associated gene sets, and 10 Phylons containing genes associated with mobile genetic elements. Six mobile Phylons were extrachromosomal, representing plasmids, and four associated with chromosomal DNA. These 31 Phylons define the structure of the *Enterobacter* pangenome. This structure is consistent with the classification of an additional 2,291 fragmented genome sequences. This structure enables the pangenome-wide mapping of genetic traits, such as motility genes, biosynthetic gene clusters, antimicrobial resistance genes, and virulence factors. NMF thus enabled phylogenetic and functional classification of genomes based on the pangenome-scale assessment of a genome's gene portfolio. A robust classification of *Enterobacter spp.* enhances the understanding of the evolution of this clinically significant pathogen.

**IMPORTANCE** *Enterobacter spp.* represent a vital member of the *Enterococcus faecium*, *Staphylococcus aureus*, *Klebsiella pneumoniae*, *Acinetobacter baumannii*, *Pseudomonas aeruginosa*, *Enterobacter* species, and *Escherichia coli* pathogens relevant for their nosocomial pathogenicity and antimicrobial resistance. Understanding the genomic diversity of the genus is vital for further study of its evolution and resistance potential. We constructed a pangenome of 777 *Enterobacter* complete genomes. Machine learning techniques were used to mathematically define major subpopulations of *Enterobacter* based on their accessory gene content, which for the first time defined dominant modes of lineage-associated and horizontal inheritance. This analysis provides insights into the distribution of traits related to antimicrobial resistance, biosynthetic gene clusters, and virulence factors. This study provides robust classification of *Enterobacter* isolates identifying differential genetic traits across the species and subspecies of the genus, overcoming some of the ambiguity in its taxonomy.

**KEYWORDS** *Enterobacter*, computational biology, genome analysis, typing, genomics, pangenomics

**Peer Reviewer** Praveen Rahi, Institut Pasteur, Phnom Penh, Cambodia

Address correspondence to Bernhard O. Palsson, bpalsson@ucsd.edu.

The authors declare no conflict of interest.

See the funding table on p. 18.

Large-scale analyses of genomes have become increasingly feasible with the rise in availability of genomic sequences across a wide range of bacterial species and strains. Such analyses can provide greater insights into the genetic structure of bacteria,

allowing for a deeper and more comprehensive understanding of their evolution, metabolism, and virulence (1, 2). In particular, pathogenic bacteria are an increasing concern to global health, with the Enterococcus faecium, Staphylococcus aureus, Klebsiella pneumoniae, Acinetobacter baumannii, Pseudomonas aeruginosa, Enterobacter species, and Escherichia coli pathogens and increasing levels of antimicrobial resistance (AMR) (3, 4) considered an urgent threat. Among these, *Enterobacter spp*. are clinically significant and present difficulties in differentiation from other gram-negative bacteria and within the species complex itself (5). *Enterobacter* is a genus of gram-negative, facultatively anaerobic bacteria that present novel challenges for treatment in the setting of nosocomial infections (6). *Enterobacter* can be found in a large variety of ecological niches, from water and soil to the human gastrointestinal tract, illustrating the widespread importance of this genus in the context of human pathogenicity and beyond (7, 8).

Over the past several decades, *Enterobacter cloacae* and other members of the species complex have been of increasing importance as a nosocomial pathogen with an evolving taxonomy (9, 10). The use of whole-genome sequences in bacterial taxonomy is an ongoing process; however, it has already served to provide advances in the evolution of bacterial species (11). The complex landscape of the taxonomy for this genus is reflected by extensive addition and reexamination of species and subspecies distinctions enabled by whole-genome sequencing (12, 13). Bacterial pangenomes, collections of the genetic content of strains across a taxonomic group, serve as a powerful methodology for resolution of traits and genetic content of a species or genus. Advancements in the availability of genomes and software techniques for their analysis have made pangenomics a vital technique for analysis of bacterial pathogens (14). The availability of genome sequences of *Enterobacter* has led to the application of pangenomic techniques that have advanced the taxonomic understanding of the genus and targeted investigations of individual *Enterobacter* species and their virulence properties (15, 16). Targeted pangenomic analysis of *Enterobacter hormaechei* and its subspecies has provided robust understanding of the pathogenicity of this species, including the notable genetic flexibility between its subspecies and the importance of plasmids in transfer of AMR and virulence mechanisms (17, 18). The scope and focus of previous research have provided significant value to the study of *Enterobacter*, and a further exploration of genetic trait distribution across the species will provide a deeper understanding of the genus at the genetic level.

Pangenomic studies have demonstrated the significance of identifying the distribution of genetic traits of interest and, through the use of machine learning methods, have enabled a deeper understanding of the pangenome structure (19, 20). Annotation of the presence or absence of genes in a given strain enables the use of machine learning techniques for the analysis of the gene distribution across strains. Matrix factorization techniques enable the calculation of an interpretable low-dimensional structure from complex biological data sets (21). Non-negative matrix factorization (NMF) is a matrix factorization algorithm that extracts non-negative components that can additively combine to represent positive input data, which can be used to represent groupings of genes and their associated strains (22). Application of this technique enables the genetic structure underlying various subpopulations of *Enterobacter* to be computed and analyzed. Accurate delineation of core and accessory genes across the genus and the roles of these genes in determination of the traits of *Enterobacter strains* can enhance the understanding of the defining genetic structure of the genus. Integration of pangenomic and machine learning methods with the growing number of available *Enterobacter* sequences has enabled increasingly accurate determinations of traits of interest and their distribution across strains of the genus, providing a more comprehensive understanding of the distribution of AMR-, metabolic-, and virulence-associated genes. In this study, we present a method that enabled analysis of dominant modes of inheritance across the *Enterobacter* pangenome through NMF. This allowed for analysis of the accessory genome including modes of both lineage-associated and horizontal gene transfer (HGT)

across *Enterobacter,* highlighting the role of both lineage-associated inheritance and HGT in the acquisition of resistance and virulence traits (23). A deeper understanding of the genetics of *Enterobacter* and the distribution of genetic traits across its different phylogroups can improve the identification of clinically important genes and their anticipated presence or absence in a given population of strains.

## RESULTS

### Data set curation

The pipeline for the generation of the *Enterobacter* pangenome and analysis of results involved downloading all available genome sequences in the public domain (see Materials and Methods). Quality control of sequences, generation of the pangenome structure, and downstream analysis of the results were performed for selection of strains for further analysis (Fig. 1a; Fig. S1). The resulting collection contained 777 complete genomes and a total of 2,291 high-quality fragmented sequences (i.e., sequences in contigs) of *Enterobacter*. Plasmids were found in the majority of strains throughout the database; however, no species possessed more plasmids than others across the genus (Fig. 1b). Across the plasmids in the data set of complete strains, 832 (43%) were predicted to be mobilizable. A total of 77 replication types were detected, with the most common being IncFIB, IncFII, and a ColE-like family of plasmids (24). Most genomes sequenced were from human hosts from a variety of isolation sites, as well as a large proportion (24%) of environmental isolates, primarily from wastewater (Fig. 1d).

Mash distance has been shown to be a valuable metric to quantify genomic sequence differences between different strains (26, 27). Pairwise Mash distances between genomes were calculated and plotted as a clustered heatmap to visualize the grouping of similar genomes across the pangenome (Fig. 1c). The largest number of strains was closely associated with *E. hormaechei* and its subspecies. *E. sichuanensis, E. pseudoroggenkampii, E. quasiroggenkampii,* and *E. chuandaensis* were the least represented members of the genus in this set of genomes, with only five genomes each (Fig. 1c; Fig. S3).

### Pangenome generation

A total of 58,421 genes were identified in the set of 777 complete genomes (Fig. 2a). Consistent with previous studies (20, 28), the genes of the pangenome were grouped into three categories: core, accessory, and rare. These categories were determined using a cumulative distribution curve-fitting method (see Materials and Methods). The *core genome* consists of genes present in all or nearly all strains (>97% of strains), containing 2,989 genes. The *accessory genome* (genes found in 7%–97% of strains in the pangenome) consisted of 4,342 genes. The *rare genome* (genes found in an individual or in a few strains [<7% of strains]) consisted of 51,090 unique genes, representing the vast majority of genes in the pangenome (Fig. 2b). The core and accessory genomes were closed, with the number of core and accessory genes quickly leveling off with increasing numbers of strains considered (Fig. S2). The rare genome was open, with the number of rare genes continuously increasing with the addition of new strains to the pangenome. A given strain had, on average, 64% of its total genes in the core genome, 27% accessory genes, and 9% rare genes across the pangenome (Fig. 2c).

### Core genome

The core genome represents a set of genes present in nearly all genomes across the genus and therefore defines many traits that are common to all *Enterobacter* species. The sole AMR gene found in the core genome was the *ampC* gene. A total of 37 genes, orthologous to virulence factors (VFs), were in the core genome, with 12 motility-related genes related to flagella representing the two largest gene functional categories. A total of 1,383 genes (43%) in the core genome were associated with metabolic functions, making up the largest category of genes in the core genome. Of the core genome, 668 (21%) genes were of unknown function (Fig. S4).

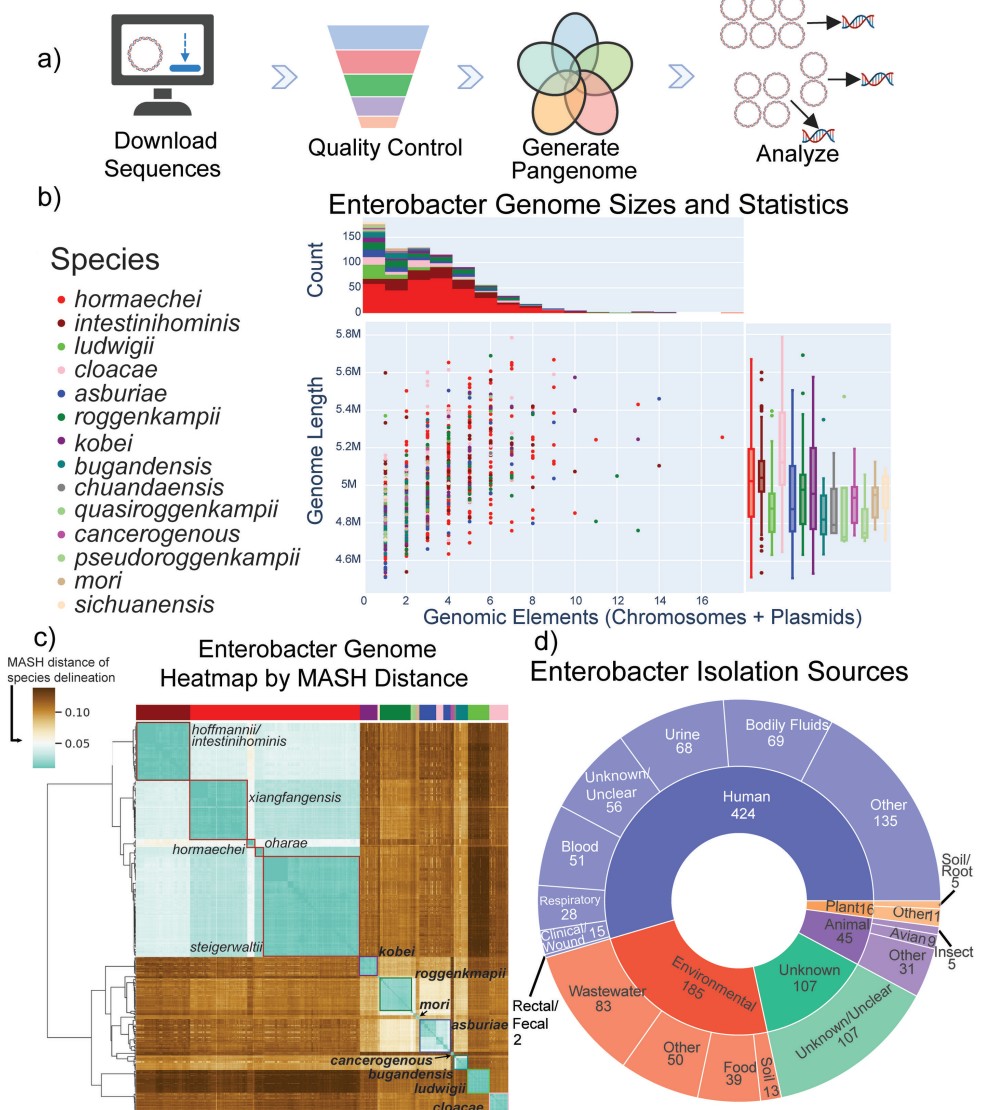

**FIG 1** Overview of the data set and pangenome generation pipeline: (a) Schematic overview of data processing and analysis steps taken in the pipeline to generate the pangenome. A total of 777 complete sequences of *Enterobacter* were selected for analysis following quality control checks (see Materials and Methods). Following construction of the pangenome, the delineation of the core, accessory, and rare genomes was performed using the protocol detailed in Materials and Methods. (b) Scatterplot displaying the number of chromosomes and plasmids on the x-axis and the genome size on the y-axis. Strains were colored with the assigned species from GTDB-Tk (25). (d) Heatmap of pairwise MASH distances for all complete *Enterobacter* genomes, annotated with species labels. A MASH distance of 0.05 represents a common distance used to distinguish between different species. There are 14 species (with five subspecies of E. hormaechei) noted. The colors for species are maintained through panels (b , c, and d). Distribution of isolation sources of strains in the pangenome. A total of 424 (54.6%) were from human sources, 185 (23.8%) were from environmental or wastewater sources, and the remaining strains were of plant (2.1%), animal (5.8%), or unknown origin (13.8%).

As core genes were present throughout the genus, the distribution of their alleles is of interest. The distribution of the number of alleles for each core gene varied greatly across the 777 complete genomes. Of the 2,989 core genes, 472 of them had one allele that was present in 50% or more of all strains in the pangenome. Of these conserved genes, 45 of these had one allele present in 95% or more of strains. These conserved genes were significantly enriched with COG category J, indicating the genus-wide conservation of RNA and ribosome-associated genes. Several COGs associated with metabolic genes, E

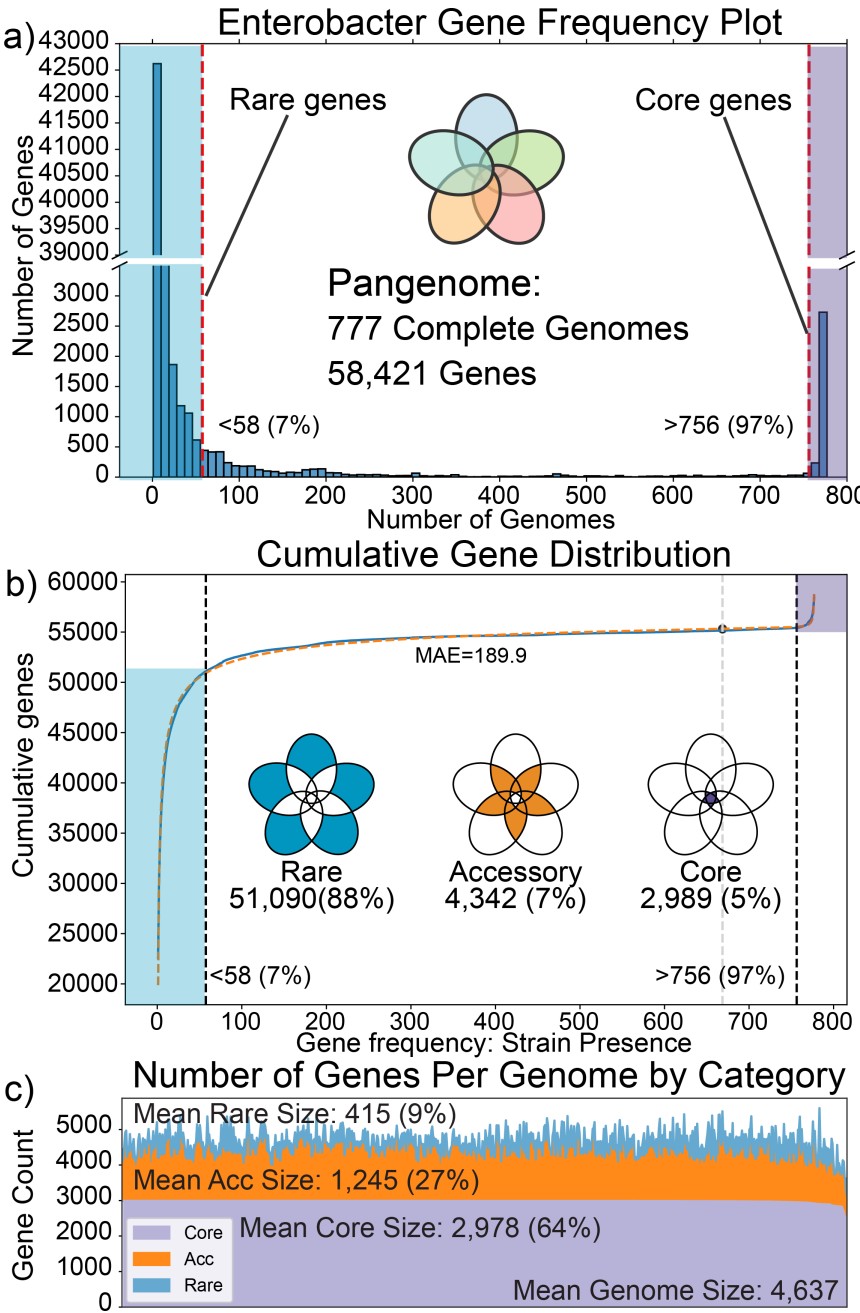

**FIG 2** Distribution of genes across the pangenome: (a) distribution of gene frequency across the 473 complete genomes in the pangenome. Genes toward the right side of the histogram are present in the majority of strains in the pangenome, with genes represented in the purple highlight being core genes, present in at least 97% of strains. Genes on the far left of the histogram represent those present in as few as a single strain. (b) Cumulative gene distribution. The gene frequency distribution was fit to a double-exponential function with a mean absolute error (MAE) of 189.9. Determination of inflection points selects the cutoffs of strain membership in order to categorize genes as core (2,989 genes), accessory (4,342 genes), and rare (51,090 genes) (details in Materials and Methods). (c) Presence of core, accessory, and rare genes in individual strains. The x-axis represents individual strains, and the y-axis is the total number of genes in each strain. The mean number of core genes in a strain is 2,978, comprising an average of 64% of the genes in any given strain of *Enterobacter*. The average strain has 1,245 accessory genes in its genome (approximately 27% of its genome), the average number of rare genes is 415, or a mean of 9% of the total genes in a strain. The strains represented on the x-axis are ordered by the number of core genes present.

and H, were significantly different between the genes, with one allele in 50% of strains and the remaining core genome, as well as COG category M (cell wall/membrane/envelope biogenesis) (Fig. S5). The conservation of core genes across the genus indicated the central function of these genes in the cellular functions of *Enterobacter*, with metabolic genes making up both the majority of core genes (43%) as well as the most allelically diverse portion of the core genome.

## NMF decomposition and analysis of the accessory genome

The accessory genome contained 4,342 genes. Of these, 48% were genes of unknown function (the largest category of accessory genes), with the second largest category being metabolic genes at 18%. The average strain had 1,245 accessory genes (Fig. 2c), representing a significant portion of the genes for a given strain and the differential gene content between major strain groupings. Therefore, analysis of the accessory genome of *Enterobacter* allows us to map differential traits of different species necessary to inhabit different niches, possess different virulence properties, and interact with hosts and the environment.

Decomposition of the accessory genome using NMF has been shown to elucidate the structure of the pangenome and identify groups of strains with similar sets of genes in the accessory genome (20). NMF decomposes the accessory **P** matrix (a genes-by-genomes matrix that only contains the rows that represent accessory genes), into two matrices such that **P = LA** (20): the **L** matrix (genes-by-Phylons) and **A** matrix (Phylons-by-genomes) (Fig. 3a). Phylons represent sets of genes that are co-occurring across strains, as defined by the **L** matrix, with the **A** matrix representing a given genome's association with each Phylon. The selection of the rank of **P** for NMF is vital to obtain the best decomposition results as the rank specifies the number of Phylons that can be found in the pangenome. Rank selection was performed by assessing the performance of the reconstruction of the pangenome **P** matrix from the NMF results, resulting in a rank of 31 being chosen as the optimal rank (Materials and Methods, Fig. S6). Thus, 31 Phylons could be identified, and their gene composition is represented by the columns of **L** (Fig. 3b).

A genome is said to have affinity for a Phylon if there is a significant weighting for that Phylon in the column of the **A** matrix. The 31 Phylons were named for the primary species or subspecies associated with them. Of the 31 Phylons, 21 represented a set of genes with affinity for only one species or subspecies of *Enterobacter*, indicating that these Phylons display accessory genes related to the strain grouping in the **L** matrix. The ten remaining Phylons were associated with strains across different species of *Enterobacter*. These Phylons displayed affinity for genes associated with plasmid-acquired genes and genes related to mobile-genetic elements. This result indicates that they represent genetic elements found to be transferred between strain groups of the genus through HGT.

The genes of the accessory genome fell into several different groups relating to the number of Phylons with which a gene was associated: 751 genes that are not present in a Phylon. The 751 genes did not offer discriminatory potential in differentiating the Phylons, while the unique Phylon genes provided evidence of a genetic signature of specific genes distinguishing the strains in a Phylon from others. Each strain was a member of one nonmobile Phylon at the most, indicating strong trait conservation within these groupings. In contrast, mobile Phylons include strains from multiple nonmobile Phylons. Notably, *mobile-1,2,3,4,6,7,* and *10* Phylons consist primarily of plasmid-associated genes, suggesting HGT as a driving force in their composition. A strain's genetic content is therefore approximated as a summation of the Phylons in which it has membership.

## Assessing the quality of the NMF decomposition

The accuracy of the decomposition was determined by using the resultant binarized **L** (named **L'**) and **A** (named **A'**) matrices to reconstruct the pangenome matrix. The

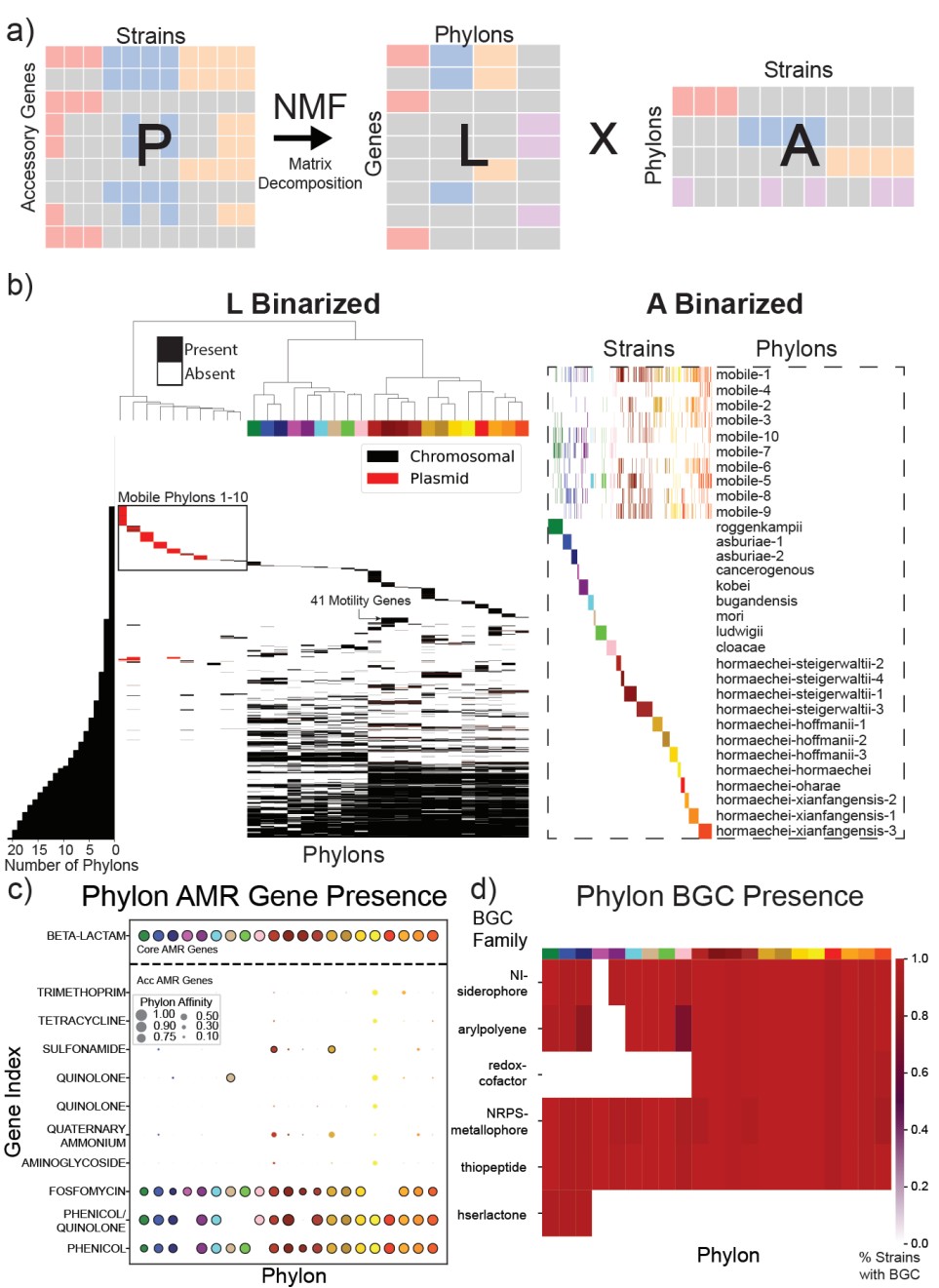

**FIG 3** Decomposition of the accessory genome using NMF: (a) Visualization of the decomposition of the accessory genome. A column of the P matrix represents the accessory genes of a given strain. A row of P represents a particular gene. An entry of 1 in a row means a gene is found in the column representing the genome, otherwise the entry in L is 0. A column in the L matrix designates a group of genes found in a series of strains, called a Phylon. The elements in a column of the A matrix denote the affinity of the Phylons to a strain when reconstituting its gene complement. (b) NMF decomposition of the accessory genome into the L and A matrices that are then binarized to visualize non-zero elements. Colors on top of the columns of the L matrix correspond to the color of the Phylon (rows) in the A matrix. The histogram to the left of the L matrix represents the cumulative number of Phylons in which a gene is present, and the rows are rank-ordered by this index. (c) The bubble plot represents the normalized affinity (between 0 and 1) of an AMR gene relating to the antibiotic class (y-axis) in a given Phylon, delineated by the colors present in Panel B. The size of a point correlates to its affinity to a given phylon, and a black border around a point indicates its presence in the associated Phylon in the binarized L matrix. All genes with at least one Phylon affinity of 0.5 or greater are plotted. (d) The presence of the six most frequently occurring biosynthetic gene clusters (BGCs) in the pangenome is displayed across the Phylons, with the percentage of strains associated with a Phylon containing a given BGC being presented in the heatmap.

reconstructed pangenome matrix ($\mathbf{P}_{rec} = \mathbf{L}'\,\mathbf{A}'$) computed 90.5% of the elements of $\mathbf{P}$ accurately. The false-positive rate was 0.01; i.e., one in 100 elements of $\mathbf{P}_{rec}$ that were determined to be absent in $\mathbf{P}$ were inaccurately computed as present. Conversely, the model has a relatively high false-negative rate of 0.30, almost entirely attributed to strains that do not binarize into a Phylon, as the Phylons do not fully capture their accessory gene content. These results showed that the binarization of the $\mathbf{L}$ and $\mathbf{A}$ matrices resulted in a conservative estimate of the gene content of strains. This indicated that the traits predicted by NMF-derived Phylons reliably correspond to the traits present in the associated strains. NMF provided a conservative yet highly accurate view of the genetic traits in the accessory genome of *Enterobacter* species and subspecies.

An accurate decomposition of the accessory genome into 31 Phylons allows us to study their differential genetic traits. We can examine the genetic composition of Phylon by Phylon to obtain a deep understanding of the structure of the *Enterobacter* pangenome.

### *E. hormaechei* strains fall into 12 Phylons across five subspecies.

Different subspecies of *E. hormaechei* had a number of different associated Phylons displaying distinguishing genetic traits. Clustering of the Phylons revealed a distinct separation of *E. hormaechei* subspecies, with 12 of the 21 species-associated Phylons belonging to *E. hormaechei*. In the accessory genome, a total of 58 genes were present across all *hormaechei* Phylons and no other Phylons, forming a "core" genome of the *hormaechei* species (Fig. 3b). Among these 58 *hormaechei* core genes, several notable traits were present. Genes responsible for synthesis of pyrroloquinoline quinone (PQQ) are present across all *hormaechei* Phylons, making up a vital part of the associated redox-cofactor BGC associated with these Phylons as well (Fig. 3d) (29). These genes provided an opportunity to differentiate the *hormaechei* species from other members of the genus (30, 31). This differentiation offered a potential insight into the use of PQQ as a cofactor for unique metabolic processes relevant to the niches associated with *E. hormaechei*.

*E. hormaechei* consists of a number of subspecies that coincide with the twelve Phylons presented here. Across all twelve of these *hormaechei* Phylons, AMR genes associated with phenicol were found, and phenicol/quinolone resistance was found in all *hormaechei* Phylons, except for *hormaechei-stigerwalittii-1*. Fosfomycin resistance genes were found in all *hormaechei* Phylons, except for *hormaechei-oharae* and *hormaechei-hormaechei*. The *hormaechei-hormaechei* Phylon possessed intermediate affinity in significantly more AMR genes than any other Phylon. Genes relating to five additional antibiotic classes had a weighting of 0.5 or greater in this Phylon (Fig. 3c).

However, these unique AMR genes were associated with plasmids. These plasmid-associated genes being in the Phylon for this subspecies displayed a potential sampling bias, with the Phylon having a limited number of strains, with many being from the same Bioproject, a study of clinical isolates in Ontario, Canada (32). However, the proclivity of strains in this Phylon to accept these AMR-associated plasmids was of note and displays the potential of this subspecies to both possess and potentially transfer multidrug-resistance phenotypes (Fig. 3c).

Three subspecies of *hormaechei* had multiple associated Phylons. Subsp. *hoffmannii* Phylons 1, 2, and 3 were distinguished primarily by the hoffmannii-1 Phylon having 165 genes, which were not present in the other two phylons. Of these genes, 26 were predicted to be metabolic genes, with 4 of these genes being associated with the *fec* operon, the acquisition of which has been associated with an increase in bacterial resistance to cefiderocol (33).

Subsp. *steigerwaltii* had four associated Phylons in the pangenome, with these Phylons having more total strains associated with them than any other group in the pangenome. Notable traits distinguishing the group included Phylons *hormaechei-steigerwaltii-1* and *4* having 41 unique genes associated with motility, primarily associated with flagellar construction. These genes were not present in any other Phylon, including

*hormaechei-steigerwaltii-2 or 3*, indicating that these additional motility genes play a key role in differentiating these strains from even closely related members of their own subspecies.

Subsp. *xiangfangensis* had three associated phylons each distinguished by sets of accessory genes unique to them. Phylons hormaechei-xiangfangensis-1 and *2* each possessed primarily uncharacterized genes distinguishing them from other phylons, with 62 and 58 genes unique to each of these respective phylons. Phylon *hormaechei-xiangfangensis-3* possessed only six unique genes; however, five of these genes were metabolic genes, four of which were associated with the phosphotransferase system, indicating a unique metabolic trait separating this population of subsp. *xiangfangensis* from the other strains.

## Notable traits of other *Enterobacter* species

The close genetic relationship between species *E. roggenkampii* and *asburiae* was displayed by their shared possession of a BGC associated with the production of homoserine lactone, with genes related to the production of this signaling molecule only associated with the roggenkampii and asburiae Phylons (34–37). These species are commonly associated with plants and plant root systems, and homoserine lactone has been shown to play a role in quorum sensing for gram-negative bacteria in plant hosts (38, 39). A BGC related to aryl polyene production was found to be associated with strains of all Phylons, except for the *cancerogenous* and *kobei* Phylons (Fig. 3d). The largest group of non-*hormaechei* strains in the pangenome was associated with the *roggenkampii* and then *asburiae* Phylons, followed by the *ludwigii* and then *cloacae* Phylons. The *cancerogenous* and *mori* Phylons represented the smallest Phylons in the number of strains, with seven and eight, respectively.

## NMF captures phylons associated with horizontal gene transfer

Ten phylons were identified to be associated with genetic traits present in strains found across different species of *Enterobacter*, indicating that these traits were related to HGT. The *mobile-1, 2, 3, 4, 6, and 10* Phylons are primarily associated with genes found on plasmids, and analysis of the sequence of plasmids across the pangenome using Mash uncovered clusters of plasmids associated with high-affinity strains for all of these, except *mobile-10* (Fig. S7). The presence of these Mash-clustered plasmids in strains corresponded to strain association with a respective mobile Phylon, indicating NMF's ability to accurately and reliably associate common HGT events with strains across the pangenome. Phylon *mobile-1*-associated plasmids were associated with a subset of IncH12A plasmids, while *mobile-2, 3, 4,* and *6* were associated with a subset of IncFIB, incL/M, IncFII/B, and IncX3 type plasmids, respectively (Fig. S7).

The *mobile-1* Phylon was found in strains across 19 different species-associated Phylons, while *mobile-9* was found across 17 species-associated phylons, *mobile-4* with five, and all others with between 10 and 16 species-associated Phylons. Genes of interest in *mobile-1* included terABCDEWYZ, genes related to tellurium resistance. Tellurium resistance genes have been found throughout numerous pathogenic bacteria and have been implicated in playing a role in oxidative stress response (40, 41); 98% of genes associated with this Phylon are related to plasmids, indicating that this is a common and widespread trait found throughout numerous different species of *Enterobacter* and exchanged horizontally between strains and potentially other bacteria in their environment (42, 43). Phylon *mobile-4* contained both the *ter* genes also present in *mobile-1* as well as genes copABCDGRS and cusABFS shared with *mobile-7*, with functions in copper resistance and ion efflux (42, 43). Phylon *mobile-6* contains plasmid-associated genes related to type IV secretion system production and conjugal transfer systems. Phylon *mobile-4* contained four AMR genes: APH(3)-Id and APH(6)-Id related to aminoglycoside resistance, *dfrA14* related to trimethoprim resistance, and *sul2* related to sulfonamide resistance. Phylon *mobile-3* possessed a *bla*$_{OXA-48}$ carbapenem resistance gene, and *mobile-6* possessed *bla*$_{NDM-1}$ carbapenem resistance and a *ble* bleomycin resistance

gene. The presence of these plasmid-associated AMR genes in the accessory genome displays frequent horizontal acquisition of AMR genes by a variety of *Enterobacter* species. NMF is able to extract Phylons representing the gene sets associated with important subpopulations of plasmids found across the pangenome, including vital resistance genes of interest.

NMF results showed significant parallels with core gene phylogeny, as well as similarity with Mash clustering results. Mash clusters and NMF results identified three and four groups, respectively, within the larger group of similar *hormaechei subsp. steigerwaltii*, displaying the ability of accessory genome-aware methods to identify meaningful genetic traits not necessarily evident from core-genome phylogenetic methods. The *pqq* operon, associated with PQQ biosynthetic genes, is present across all *E. hormaechei* Phylons but absent in all other species and Phylons. Three accessory genes in the pangenome were detected as Salmochelin-associated. All strains containing all of these genes were annotated (Fig. 4). These genes were found throughout strains of all three *hormaechei-steigerwaltii* Phylons, as well as *E. cancerogenous* and a subpopulation of strains in *E. bugandensis*. The *arn* operon, associated with resistance to colistin, was not detected in the Phylons for *hormaechei-steigerwaltii-4,* any *hormaechei-xiangfangensis* Phylons, *hormaechei-oharae, hormaechei-hormaechei,* or the *hormaechei-hoffmannii Phylons.*

Strains with multiple genes labeled as members of the *arn* operon are annotated, showing high consistency with the Phylon structure and no strains in unassociated Phylons possessing these genes (Fig. 4). Associations between Mash clusters and characterized Phylons also display a clear structure, with most Phylons displaying a direct correspondence with a single Mash cluster or a combination of two (Fig. S8). The *asburiae-1* and *2* Phylons consist of strains from three Mash clusters each, indicating that they captured the shared traits between these subgroups of E. *asburiae*. Similarly, the composition of the *roggenkampii* Phylon corresponds to two Mash clusters. Mash cluster 12 maps to strains associated with the *hormaechei-stiegerwaltii-1* and *3* phylons, while some strains associated with cluster 12 also remain unclassified by the Phylon structure, indicating that whole- and core-genome similarities between these strains did not reflect similarities in the accessory genome that NMF was able to capture or significantly distinguish as a unique Phylon.

Thus, the Phylon structure obtained through NMF decomposition of the accessory genome gives a detailed definition of the gene composition of the subgroups of *Enterobacter* strains. In fact, the NMF decomposition offers a genome-scale definition of the differential gene contents of the Phylons and thus the genetic basis for differential traits between them.

## Rare genome

The rare genome consists of 51,090 genes, 68% of which are poorly characterized (Fig. S4). Of the 214 total AMR genes found in the pangenome, a total of 170 (79%) were found in the rare genome. Similarly, 147 (67%) of the 218 VFs were also found in the rare genome, displaying acquisition of diverse traits related to pathogenicity across *Enterobacter* strains. Analysis of these rare genes, while they are not present in all strains, is vital to understanding potential avenues for pathogenicity and antibiotic resistance, which may be further integrated by populations of *Enterobacter*.

Associations between the Phylon structure and these traits also display potential avenues for understanding the pathogenicity of different species and subspecies of the genus. The number of AMR genes per strain is highly variable, with *hormaechei-steigerwaltii-2/4, hormaechei-hoffmannii-1,* and *hormaechei-xiangfangensis-3* having the highest average number of rare AMR genes per strain. Notably, a multidrug efflux transporter *mexF* was found to be near-ubiquitous across strains of the *E. ludwigii* in the pangenome, indicating that this gene and AMR trait appears to be a conserved genetic trait possessed by the species. Similarly, an orthologous gene cluster of *mexF* was found across all strains

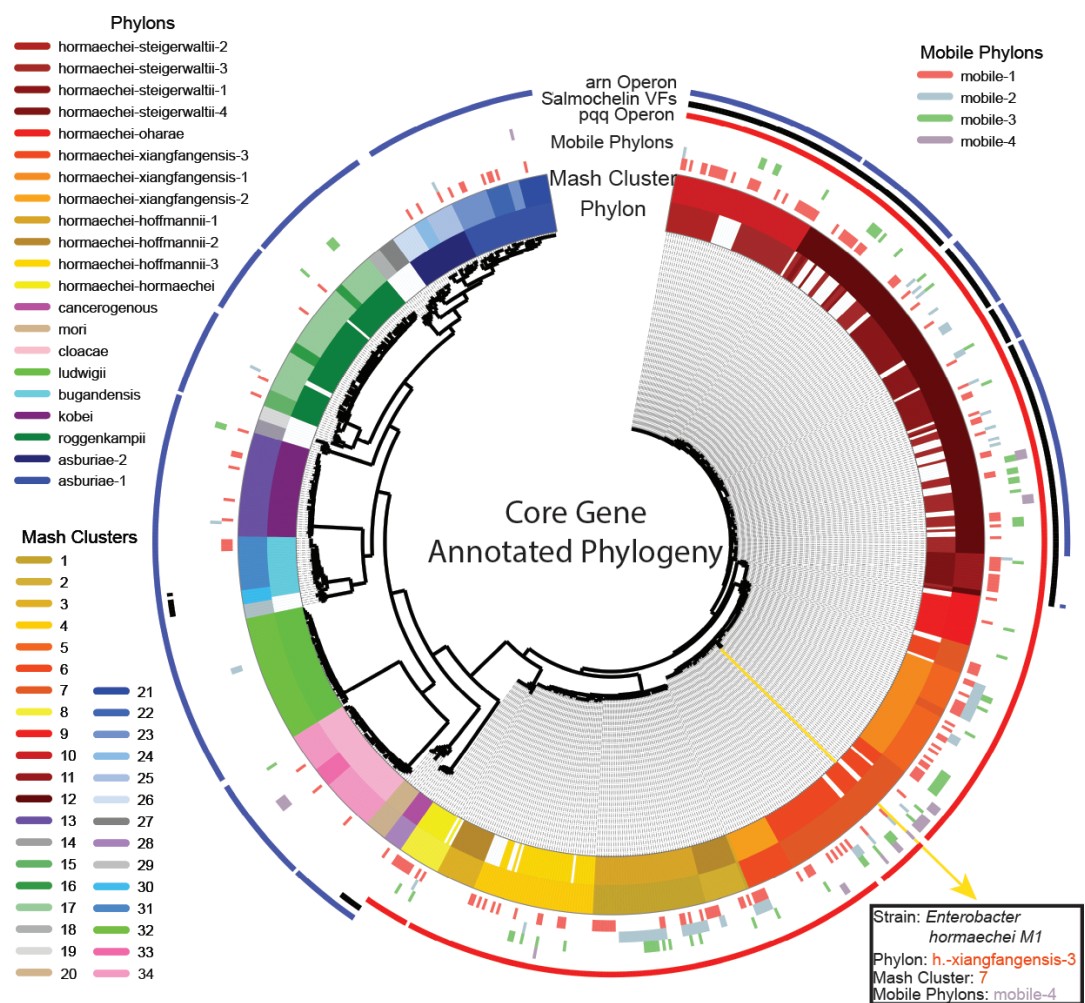

**FIG 4** Core gene phylogeny of the *Enterobacter* pangenome colored by strain groupings (by Mash distance or Phylon): core genes across the pangenome were used for construction of the phylogenetic tree using MAFFT alignment of the concatenated genes and FastTree to construct the phylogenetic tree structure. Phylon classification of strains is displayed on the inside ring, while fine-grained Mash clusters of strains are displayed on the second ring (see details in Fig. S3). Four rings are present to represent the distribution of mobile Phylons across the strains in the pangenome. The outermost rings display the distribution of several traits of interest associated with the Phylon structure: the presence of an arn operon, genes associated with salmochelin virulence factors, and presence of a PQQ operon. The yellow line radiating from the center to the bottom right represents a single strain in the phylogeny with the associated annotations for a given strain.

with affinity for the *mori* Phylon. Additional sampling of *ludwigii* and *mori* strains may result in these genes becoming a part of the accessory genome (Fig. 5a).

Analysis of the location of AMR and VFs was performed to discover both known and unknown pathogenicity islands (Fig. 5b). The majority of rare AMR and VF genes are found as singletons in the pangenome, not colocated with any other rare virulence traits. However, instances of co-possession of these traits can be seen in several cases of interest, mainly related to VFs. A yersiniabactin pathogenicity island, a siderophore associated with pathogenic *Enterobacteriales*, (44, 45) was found across 21 strains. These strains fall into six different Phylons and in isolates from 2007 to 2022 in four distinct geographic regions (Fig. 5c; Fig. S9). The rare genome of *Enterobacter*, though poorly characterized, contains many AMR and VFs, revealing a diversity of strain-specific traits and conserved resistance elements like *mexF*, as well as distinct pathogenicity islands across diverse strains.

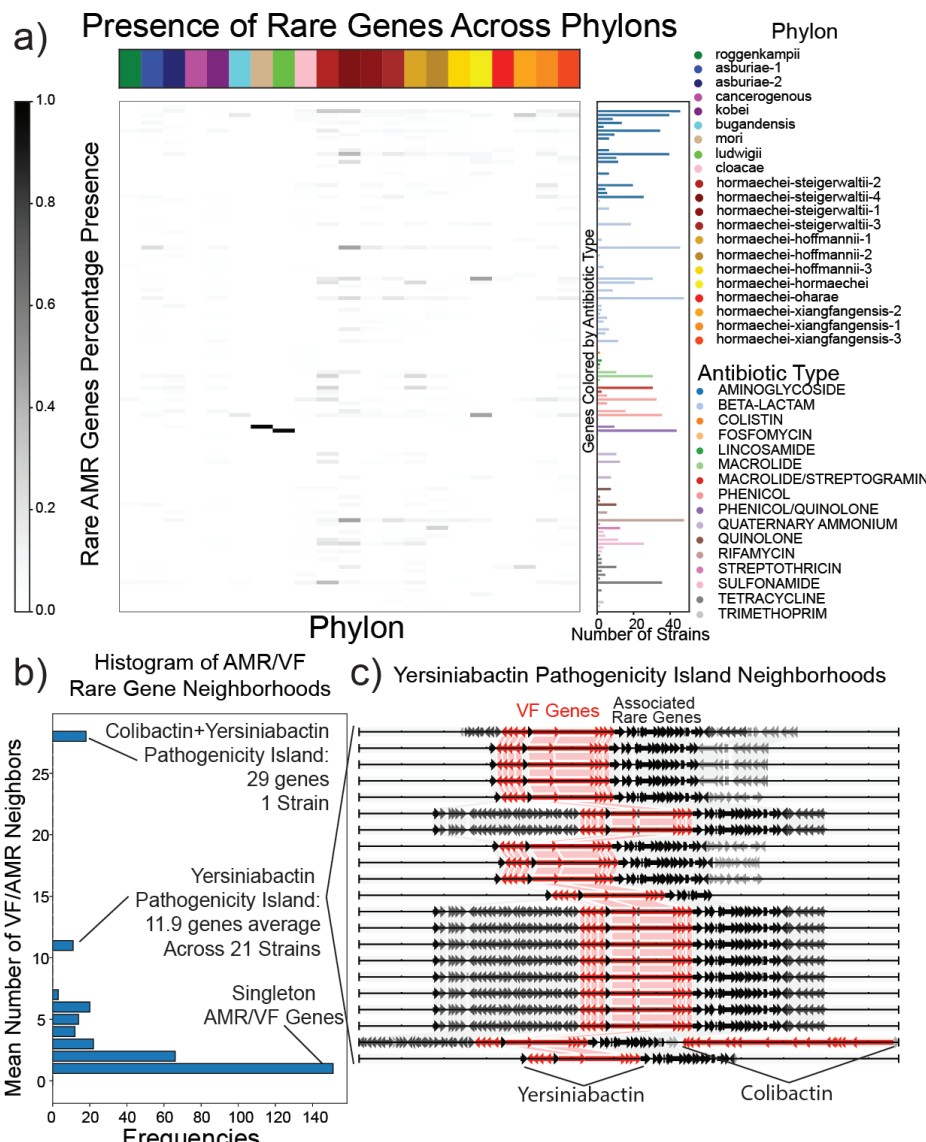

**FIG 5** Rare genome analysis of AMR and VF genes relative to the Phylon structure: (a) presence and distribution of AMR genes across the rare genome and the Phylons. The heatmap displays the percentage of strains in a given Phylon that possess a specific AMR gene across the 147 computationally detected AMR genes. The bar plot on the right side represents the number of occurrences of a given gene in the pangenome. The most frequent classes of AMR genes present in the rare genome are beta-lactams and aminoglycosides, with 37 and 25 different rare genes, respectively. The class with the most occurrences is aminoglycosides, with 349 total occurrences of aminoglycoside AMR genes across 25 different genes. (b) The neighborhood of rare genes throughout the pangenome was calculated by determining the total number of rare genes found in a contiguous region in a genome, uninterrupted by core or accessory genes. The mean number of neighbors for a gene includes a mean of all rare genes found in a contiguous region for a given gene in each genome. Histogram of the mean number of AMR/VF neighbors for a gene. The majority of AMR and VF genes are singletons, having no other AMR/VF neighbors and being found in only one strain. There are 11.9 VF genes on average associated with yersiniabactin production found as neighbors in 21 strains containing the genes related to the production of yersiniabactin. (c) Genome diagram of the rare-gene neighborhood of the 21 strains containing the yersiniabactin pathogenicity island. Nine genes related to yersiniabactin production are found in the same orientation in all strains; however, ten have a small hypothetical protein inserted in the middle of the pathogenicity island. One strain also possesses genes related to colibactin production in the same region of rare genes. The red arrows represent VF genes, while gray and black genes represent other rare genes. Black genes indicate conservation of the gene across all nine instances of this pathogenicity island. The links between genome tracks display links between the same rare genes across adjacently displayed genomes. The strains possessing these rare genes are found in a total of five different Phylons, from a total of five different countries and isolates ranging from 2013 to 2022 (Fig. S9).

## Affinity of whole-genome sequences in contigs to Phylons

Affinities of the incomplete (i.e., sequence is fragmented in the form of contigs) genome sequences outside of the pangenome to the calculated Phylons were determined based on their similarity of gene presence and absence to the structure of the **L** matrix determined by NMF (see Materials and Methods). A total of 2,291 genomes available in contigs were analyzed and assigned to Phylons, with 1,692 strains being assigned to a species-associated Phylon (Fig. S10). The majority of strains were classified as *hormaechei*, specifically into the *hormaechei-xiangfangensis-2, hormaechei-steigerwaltii-3,* and *hormaechei-hoffmannii-1. Roggenkappii and kobei* were the most common non-*hormaechei* Phylon, while *mori* and *cancerogenous* were the least common Phylons found in the population of the sequences analyzed. Mobile Phylons 1, 8, and 9 were all common throughout the population of sequences, with mobile-1 being most common.

Thus, we show that the Phylon structure deciphered from the 777 complete genomes is stable, and it applies to other genomes whose sequence is incomplete and represented by contigs. Scripts to analyze new sequences and assign them affinities to the Phylon structure presented here, as well as assignment to the Mash clusters presented here, are available (see Materials and Methods).

## DISCUSSION

*Enterobacter spp.* are nosocomial pathogens displaying an ability to resist antibiotic treatment through AMR genes. *Enterobacter spp.* is considered to be an urgent threat. This study applies NMF to present a novel decomposition of the pangenome matrix of the *Enterobacter* genus. Based on 777 complete genome sequences of strains of the genus, the main results of this work are as follows: (i) determination of the core, accessory, and rare genomes of the *Enterobacter* pangenome; (ii) decomposition of the accessory genome into 31 Phylons, each of which is defined by a unique groups of genes; (iii) deconvolution of the modes of horizontal- and lineage-associated inheritance across the pangenome and how key traits fall across the genus.

*Enterobacter* represents a diverse genera of several pathogenic species, regularly grouped together in scientific studies and treatment (6, 29). Its genetic diversity necessitates a deeper understanding of the traits that are conserved between and differentiate *Enterobacter* species and their subspecies. The boundaries of the core, accessory, and rare genomes (composed of 2,989, 4,342, and 51,090 genes, respectively) were calculated. This allowed for a robust and specific understanding of genes and associated traits that are common between *Enterobacter* species and that are unique to specific species, subspecies, or strains. The core genome represents genes shared across the genus. The diverse rare genome provides information on traits that are not commonly present but display the wide variety of genes available to contribute to the fitness of the genus. The accessory genome, however, provides significant information in defining and characterizing subpopulations within a pangenome (46).

The use of NMF for decomposition of the accessory genome has demonstrated success in determining meaningful subpopulations of bacterial species (20). This study presents a novel genus-level decomposition of *Enterobacter* into NMF-calculated Phylons. These Phylons provide 31 sets of genes in the pangenome that define the accessory genomes of *Enterobacter* strains. Highlighted by NMF are Phylons that parallel the existing taxonomic structure of the best studied species of *Enterobacter*, with twelve Phylons found to characterize the *hormaechei* species. Complete genomes were selected for the purposes of this analysis in order to maximize the quality of the data included for this approach, which may introduce sampling bias as compared to the complete set of available genomes but ensures a high standard of quality of the input data for this method.

Each Phylon provides a set of genes that define the accessory genome of the strains associated with it, providing a basis for the genetic content of the strains. Several Phylons represent horizontally acquired genes, displaying both common and diverse patterns of

acquisition of the horizontally acquired genetic material by *Enterobacter spp.* as well as the ability of the methods presented herein to determine these modes of acquisition within a pangenome. The combined genetic content of the Phylons associated with a given strain provides distinguishing information that can be used to define and determine populations of interest within *Enterobacter*.

This study also highlights the importance of utilizing genome-scale context for robust understanding of the underlying genetic basis for the differentiation of different species of *Enterobacter*. Whole-genome sequencing allows for a more robust identification of isolates as compared to tools such as MLST typing and *hsp60* typing (12, 47). This study presents a framework for associating new isolates of *Enterobacter* species into Phylons and highlighting their associated genetic traits, helping increase understanding of *Enterobacter* genetics and population structure. The Phylon structure, as determined by the accessory genome, also displayed clear parallels to whole-genome similarity methods as well as core-genome phylogeny while providing additional information regarding accessory genome content. Use of whole-genome and accessory genome content for analysis of phylogeny and strain typing provides more accurate and informative results in identification of strains and their underlying genetic composition for further study and for treatment.

Further study of *Enterobacter spp.* is of vital importance given the continued relevance of the genus as a nosocomial pathogen and widespread environmental organisms. Among *Enterobacter* species, *E. hormaechei* stands out as the most notable pathogen in the genus (7). There are numerous genetic traits of interest associated with addressing this threat, present across the species as a whole and in specific subpopulations and subspecies. Of note is the increased presence of motility-associated genes in subpopulations of *E. hormaechei* subsp. steigerwalttii and the high affinity of plasmid-associated AMR genes in *Enterobacter hormaechei* subsp. hormaechei, highlighting the pathogenic potential of key subpopulations. Additionally, the acquisition of a PQQ operon by *E. hormaechei* indicates the importance of this metabolite to the fitness of the species as members of rhizospheric communities and the association of this metabolite with numerous opportunistic pathogens (48, 49). These findings reveal the importance of the accessory genome in shaping the clinical and ecological significance of *E. hormaechei*.

With the expansion of whole-genome sequencing, utilization of available sequencing data to further understand the population genomics of organisms of interest is paramount. As more high-quality genomes of *Enterobacter* and other nosocomial pathogens of interest continue to be sequenced, a deeper understanding of the underlying genetic basis of undersampled groups will be obtained, as well as a more precise definition of the conserved genetics of more commonly studied species. These advancements will provide a further understanding of this organism, which is a vital member of microbial communities in human, animal, and plant microbiomes, as well as a pathogen with relevance to AMR gene acquisition and transfer.

Taken together, the results show that the accessory genome provides vital information to classify genetic traits and phylogeny of the *Enterobacter* genus. The use of NMF for the calculation of Phylons can deconvolute elements of lineage-associated inheritance and horizontally transferred genes. HGT plays a vital role in augmenting fitness of strains within the genus, with resistance traits being horizontally acquired through plasmids across numerous different species of *Enterobacter*. These insights highlight the potential of machine learning approaches such as NMF to unravel complex biological processes, shedding light on inheritance patterns and trait distribution in *Enterobacter*.

## MATERIALS AND METHODS

### Collection and processing of sequences from BV-BRC

All genomes from BV-BRC (50, 51) labeled as *Enterobacter* were considered and assessed for quality from October 2023 and earlier (Fig. S1). Sequences were filtered to have an

N50 score greater than 3,780,000 and an L50 score of 1 for complete sequences. The N50 cutoff was determined using 90% of the N50 score for the NCBI (52) reference genome of *E. hormaechei* (assembly ASM1904824v1) (Fig. S11a). Genomes with more than 282 contigs were removed based on previous standards for removal for contig counts (28). Genomes with CheckM contamination < 2.7% or Completeness < 97.4% were removed. These thresholds for completeness and contamination were determined by determining the elbow and knee of the data set, respectively, in order to filter outliers for contamination. Strains with fewer than 4 MB in genome length or fewer than 1,000 predicted Patrick CDSs were not considered. GC content was limited to between 54% and 57% as most Enterobacter species possess GC content between 54.5% and 56% and the above limits were selected for removal of outliers (Fig. S11b). Accessions with more than 20 annotated plasmids were removed as outliers relative to the number of plasmids found in other genomes in the data set. Further filtration was performed by deduplication of sequences and enforcing maximal Mash distance in the pangenome (described later). Complete genomes were used for development of the NMF model and Phylon structure. An additional 519 complete sequences in addition to those gathered from BV-BRC were processed from NCBI Genomes Database (as of August 2025) and were subjected to filtration criteria.

## Genome annotation and pangenome creation

Genomes annotation and reannotation were performed using Bakta (53). The pangenome of coding sequences was constructed using Panaroo v1.5.2 (54). The core genome threshold was initially determined using CD-HIT to construct an initial pangenome structure. Sequences were clustered using CD-HIT, and a core genome threshold of 0.965 was established based on the thresholds established by construction of an initial pangenome using CD-HIT (determination of core genome size described later in the Materials and Methods section) (55, 56). For CD-HIT, cutoffs of 80% amino acid sequence similarity and 80% sequence alignment coverage were used, in line with previous studies (20). Panaroo was run using the options "--clean-mode sensitive," a family threshold of 0.8 in line with the CD-HIT protocol, and the core threshold set based on the above approximation from CD-HIT. The "--remove-invalid-genes" option was used in order to enable the Bakta-annotated files to be used as input for Panaroo. Representative sequences from the pangenome were selected as the longest sequence without premature stop codons from each gene cluster computed by Panaroo. These were then annotated using eggNOG, AMRFinder, and VFDB (57–60). For annotation of sequences from VFDB, a blast was performed between all representative sequences across the pangenome and the amino acid sequence database from VFDB. Genes with an E value < $e^{-10}$ and similarity > 80% were annotated as VFs, with the highest similarity annotation being selected if multiple significant alignments occurred for a given reference sequence in the pangenome.

BGCs were computed using BGCFlow (61) and antiSMASH (62). All genomes were annotated using BGCFlow, and BGC families were determined based on shared family ID from the pipeline. Taxonomic assignment from the Genome Taxonomy Database (GTDB) was assigned using the GTDB Toolkit (GTDB-Tk) classification functionality in order to ensure up-to-date taxonomic information for genomes included in the pangenome (25, 63). Plasmid typing was performed using MOB-Suite's (version 3.1.9) MOB-typer functionality (64).

## Mash distance calculation and sequence filtering

Pairwise Mash distances (26) were calculated for all sequences in the pangenome. The Mash distance calculation command was run using kmers of size 21 and a sketch size of 1,000. Sequences with Mash distance greater than 0.205 being removed as being in the top 1% of farthest distance from the mean of representative sequences for *Enterobacter* species present in the pangenome (representative sequences based on the *Enterobacter*

genus listed on BV-BRC). Mash distances were converted into Pearson correlation distance values, as described in previous studies (27), and were then clustered. Sensitivity analysis was performed in order to optimize the threshold for clustering using Ward's minimum variance, resulting in a value of 0.25 for thresholding the hierarchical clustering (Fig. S12). Seaborn's (65) clustermap function was utilized to generate the clustering plot, with clusters determined by Ward's minimum variance. Iterative removal of clusters with sufficiently underrepresented and disparate genomes (fewer than five members in a cluster) was removed in order to ensure sufficient representation of sequences for downstream application. Mash clusters were then annotated based on the taxonomic assignment given to genomes within the cluster by GTDB-Tk (25). A total of 34 Mash clusters were determined and annotations computed (Fig. S3).

## Determination of the core, accessory, and rare genomes

The core, accessory, and rare genomes were defined based on the cumulative distribution plot (Fig. 2b) using methods described in the previous work (20, 28). Determination of the inflection point of the data through curve fitting to an S-shaped curve was performed, and the core genome was defined as being 90% of the way from this inflection to the presence in all strains, resulting in genes with > 97% (756 strains) presence being in the core genome. From the inflection point of this curve, 90% of the distance from the inflection point to presence in no strains was used to determine the boundary of the rare genome, resulting in genes with < 7% (58 strains) presence being placed into the rare genome.

## Non-negative matrix factorization

NMF was run through the Scikit-Learn implementation (66) to calculate the decomposition. The methodology of NMF calculation was based on previous work (20). Rank was determined by running NMF across ranks one through 99 and computing the accuracy metrics of each, including Bayesian information criterion, residual sum of squares, and Akaike information criterion (AIC). The final rank of 31 was selected as it was determined to be the knee, or point of diminishing returns, for increasing the rank in lowering the AIC. (Fig. S6). Following this, NMF was run 50 times at this rank in order to select the best run for downstream analysis and characterization of Phylons. The L and A matrices were binarized in order to increase the interpretability of the results and define the genes making up each phylon. This was performed using 3-means clustering in scikit-learn. Columns of L were clustered into three groups, and the cluster with the highest mean value was binarized to 1, while other clusters were binarized to 0. The same procedure was used to binarize the rows of A, with this method also having been described in previous work (20). For visualization of the **L** and **A** matrices, clustering of the columns of L was performed using Ward's minimum variance method, and this order of columns was preserved in the ordering of the rows in the visualization of the A matrix. The color of a gene in the L matrix indicates if it is primarily (more than 50% of occurrences) found on the chromosome of the strains in the pangenome or on plasmids.

## Analysis of Phylons

Phylons named after *Enterobacter* species and subspecies were named based on their associated Mash clusters and genomes, with 21 Phylons being associated with lineage-associated genes. A Phylon was associated with a Mash cluster if it had strains in it (binarized to 1 in the A matrix) which were present in a given Mash cluster (Fig. S8). Mobile Phylons were determined as Phylons with strains found across multiple distinct species and subspecies, as well as possessing significantly fewer genes than in the taxonomic Phylons. Presence of genes associated with HGT leads to the naming of these Phylons as mobile and mobile genetic element (MGE)-associated. Notably, no strain was present in more than one Phylon determined to be taxonomic, while strains were found to be associated with multiple mobile Phylons. BGC associated with a Phylon was

determined by the percentage of strains in a Phylon, which were annotated as having a BGC of the given family.

## Creation of phylogenetic trees

The phylogenetic tree generated was based on the core genes of the pangenome. Genes were concatenated and then aligned using command line MAFFT (67) (version 7.525) using default parameters. FastTree (version 2.1.11) was used to construct the tree using a generalized time-reversible model (68). The resulting tree was visualized with PyCirclize (69) (version 1.4.0).

## Rare genome analysis

The co-location of rare genes of interest was determined by determining if rare genes were found in a contiguous region in any genome in which they were present, that being if two rare genes were part of a region of rare genes that was uninterrupted by core or accessory genes in a genome. Visualization of the yersiniabactin pathogenicity islands found in the pangenome was done using BioPython's GenomeDiagram module (70).

## Affinity calculation of whole-genome sequences to Phylons

A total of 2,291 high-quality non-complete whole-genome sequences (WGS) were downloaded from BV-BRC and gene membership to the pangenome gene clusters calculated from the complete sequences using CD-HIT-2D with an 80% sequence similarity threshold in order to generate a $P_{WGS}$ matrix for the accessory genes. Following this, the normalized $L$ matrix from the generated Phylon structure was used as input in addition to the $P_{WGS}$ matrix in order to calculate the $A_{WGS}$ matrix, which contains the affinities of the WGS sequences to the Phylons. This was done using non-negative least squares implementation in SciPy (71, 72). Affinities for strains to Phylons were binarized using thresholds determined using three-means clustering for binarization of the $A_{WGS}$ matrix for the inferred Phylon structure. This resulted in no strain being assigned to more than one characterized Phylon in the $A_{WGS}$ matrix, consistent with the structure of the $A$ matrix for the Phylon structure calculated in this work.

### ACKNOWLEDGMENTS

This work was funded by the Novo Nordisk Foundation (Grant Number: NNF20CC0035580).

Conceptualization: B.O.P., J.T.B., and S.M.C., Data curation: J.T.B., Investigation: J.T.B., S.M.C., and G.L., Methodology: J.T.B., S.M.C., and J.M.M., Mentorship: J.M.M. and B.O.P., and Writing and editing: All authors.

### AUTHOR AFFILIATIONS

[1]Department of Bioengineering, University of California San Diego, La Jolla, California, USA

[2]Department of Pediatrics, University of California San Diego, La Jolla, California, USA

[3]Bioinformatics and Systems Biology Program, University of California San Diego, La Jolla, California, USA

[4]Center for Microbiome Innovation, University of California San Diego, La Jolla, California, USA

[5]Novo Nordisk Foundation Center for Biosustainability, Technical University of Denmark, Lyngby, Denmark

### AUTHOR ORCIDs

Joshua T. Burrows http://orcid.org/0000-0001-6595-8736

Jonathan M. Monk [ID] http://orcid.org/0000-0002-3895-8949
Siddharth M. Chauhan [ID] http://orcid.org/0000-0001-6674-895X
Bernhard O. Palsson [ID] http://orcid.org/0000-0003-2357-6785

## FUNDING

| Funder | Grant(s) | Author(s) |
|---|---|---|
| Novo Nordisk Fonden | NNF20CC0035580 | Bernhard O. Palsson |

## AUTHOR CONTRIBUTIONS

Joshua T. Burrows, Conceptualization, Data curation, Formal analysis, Investigation, Methodology, Software, Validation, Visualization, Writing – original draft, Writing – review and editing | Gaoyuan Li, Investigation, Software, Writing – review and editing | Jonathan M. Monk, Methodology, Supervision, Writing – review and editing | Siddharth M. Chauhan, Conceptualization, Investigation, Methodology, Software, Supervision, Writing – review and editing | Bernhard O. Palsson, Conceptualization, Funding acquisition, Project administration, Resources, Supervision, Writing – review and editing

## DATA AVAILABILITY

Code for construction of the pangenome and generation of figure panels, as well as the ability for users to construct a pangenome and Phylon structure including strains of interest to them, are available at https://github.com/jtburrows/Enterobacter_pangenome. Data necessary for analysis of the Phylon structure are available at https://doi.org/10.5281/zenodo.15595723.

## ADDITIONAL FILES

The following material is available online.

### Supplemental Material

**Supplemental Figures (Spectrum01922-25-S0001.docx).** Fig. S1 to S12.
**Supplemental Tables (Spectrum01922-25-S0002.xlsx).** Strain and gene metadata as well as the L and A binarized matrices.

### Open Peer Review

**PEER REVIEW HISTORY (review-history.pdf).** An accounting of the reviewer comments and feedback.

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
