## [Reviewer comments · Microbiology Spectrum]

Microbiology Spectrum

Structure of the Enterobacter Pan-Genome is Revealed Using Machine Learning

Joshua Burrows, Gaoyuan Li, Jonathan Monk, Siddharth Chauhan, and Bernhard Palsson

Corresponding Author(s): Bernhard Palsson, University of California San Diego Jacobs School of Engineering

Review Timeline:

Submission Date:	June 23, 2025
Editorial Decision:	August 29, 2025
Revision Received:	October 28, 2025
Accepted:	November 16, 2025

Editor: Angela Re

Reviewer(s): Disclosure of reviewer identity is with reference to reviewer comments included in decision letter(s). The following individuals involved in review of your submission have agreed to reveal their identity: Praveen Rahi (Reviewer #2)

Transaction Report:

DOI: <https://doi.org/10.1128/spectrum.01922-25>

Re: Spectrum01922-25 (Structure of the Enterobacter Pan-Genome is Revealed Using Machine Learning)

Dear Prof. Bernhard O Palsson:

Thank you for the privilege of reviewing your work. The manuscript in the current form is not acceptable for publication. For your knowledge, below you can find the revisions provided by the reviewers and some comments of mine. Should you still deem it of your interest to submit the revised version of the manuscript to Microbiology Spectrum, please proceed to a point-by-point careful and substantial revision of the work by including necessarily the revision of both the input datasets and of the analysis underlying the manuscript.

Below you will find my comments, instructions from the Spectrum editorial office, and the reviewer comments.

Editor's comment

Introduction

- Authors are invited to specify which are the elements of methodological novelty brought about by this work. For instance, they could reassess the methodological comparison with their own refs. 14-15 and appropriately modify the manuscript.
- Lines 55-63. Which are the taxonomic advances repeatedly evoked by the authors? The authors are invited to detail their statements as they are relevant to capture the significance of the proposed work.
- Lines 70-80 enumerate a sequence of potential future benefits coming from the application of pangenome analysis tools. However, the Introduction section be focused on the present manuscript content.
- Acronyms should be described at the first instance (ESKAPEE)

- Results - Dataset curation
- Lines 90: Please rephrase "no species appeared to be overrepresented in plasmid possession"
- Figure 1 legend. Please, rephrase "detailed previously (see Methods)" as the Method section follows Figure 1. Please, correct the numbers reported in the legend as they are not consistent with the numbers displayed in Figure 1b. Figure 1c: Please, improve the visibility of strain colors. Figure 1d: "distinct" is an adjective whereas a verb is needed. Authors state "There are 9 species (with 5 subspecies of *E. hormaechei*) noted" but the figure does not show the subspecies. Figure 1e: Please, rephrase "Treemap of species distribution of dataset based on annotated species on BV-BRC." Authors are invited to remove Fig. 1e that is superfluous and not cited in the manuscript. Authors could place it in the Supplementary Material. Please, increase parallelly the readability of the heatmap in pane d).
- How many clusters were obtained by CD-HIT?
- If the core genome contains genes present in more than 97% of strains, this information is in contrast with Line 157 do the authors state that 430 genes are said to have one allele present in 50% or more of strains.
- What is the functional annotation source of the pangenome?
- Do the authors consider only the representative genomes of MASH-based clusters in NMF? Please, clarify lines 205-207 and explain clearly which are the genome involved at the beginning of the section as it can improve readability.
- Lines 219-224 let me think that phylons contain strains whereas they are made of genes. Please, clarify.
- How are false positives and false negatives computed in "Assessing the quality of the NMF decomposition"?

Methods

- N50 and L50 threshold. Please, provide references/justifications for the thresholds used.
- CheckM contamination <2.7% or Completeness <97.4%. Please, provide justifications.
- Please, provide reference for Patrick. How was it used?
- Please, provide justifications for the allowed range of GC content
- Please, rephrase "as well as accessions with more than 20 annotated plasmids being removed as outliers"
- Authors must specify which genomic features provided by Bakta (coding, non-coding and so on so forth) are used in the subsequent analysis.
- Line 511-512. Authors are invited to rephrase the sentence and to specify the alignment percentage minimum cutoffs since citations are not sufficient.
- How do the authors select the representative sequences in CD-HIT?
- The BGC annotation is not clear. What is the pangenome which the authors refer to here? Please, clarify the insufficient description provided in lines 519-522.
- Lines 527. What does "from BV-BRC annotation." mean?
- Lines 528-520. Sensitivity analysis was performed to optimize the threshold for clustering, resulting in a value of .15. Authors are invited to provide evidence (plots, tables) of this analysis.
- Is 0.15 a Pearson's correlation coefficient (PCC=)?
- Why do the authors transform a MASH distance into a PCC and then the PCC into a distance?
- Authors must clarify which are the strains used to annotate the MAS-based clusters? Lines 531-535 are unclear. What do the authors mean by "type strain"?
- Authors are invited to correct the definition of rare genes at lines 541-543

- At lines 545-561 the choice of the rank range and the choice of the final rank in non-negative matrix factorization are poorly described. What are the accuracy metrics mentioned? How are they used jointly with AIC? Authors are invited to explain the reason underlying binarization. The intermittent references to the previous article of the authors turned out inappropriate. Even though NMF is widely used, it is useful to introduce the formula linking L and A.

Revision Guidelines

Sincerely,
Angela Re
Editor
Microbiology Spectrum

Reviewer #2 (Comments for the Author):

The manuscript "Structure of the Enterobacter Pan-Genome is Revealed Using Machine Learning" demonstrates that NMF-based accessory genome clustering, "Phylon", provides a powerful, most likely reproducible, and biologically meaningful approach to classify Enterobacter species and subspecies, particularly for challenging groups such as *E. hormaechei*. It provides genome-scale, trait-based definitions that can reconcile or refine ambiguous or contentious taxonomic groupings. However, the manuscript lacks several key points, please the attachment.

Reviewer #3 (Comments for the Author):

This manuscript provides an important analysis of the pan-genome of Enterobacter. Particularly, the authors use machine learning (non-negative matrix factorization) to identify clusters of accessory genes that best represent sub-groups of

Enterobacter.

The paper is clearly written. I do appreciate the effort to provide biological interpretations for the different groups of accessory genes, especially on how they relate to AMR, virulence, and biochemical pathways.

Source data and methodologies are well-described and publicly available via Github.

Altogether, this will be an important resource for microbiologists and evolutionary biologists.

Very minor comments:

*What is the justification for the 7-97% cutoff on the definition of accessory genes?

*Is the result of NMF decomposition unique? How robust is this to the cut-off definition above?

Reviewer #4 (Public repository details (Required)):

The authors must provide the list of genes that constitute each "phylon." Without proving this information, we are just taking their word that this approach any useful biological meaning over the vast plethora of prior pan-genome knowledge.

Reviewer #4 (Comments for the Author):

This paper describes the pan-genome of *Enterobacter* spp. using a machine learning approach; however, there are several major flaws with this paper that prevent it from being acceptable in mSpectrum.

- 1) What is a pan-genome (I know what it is, but many microbiologists that read this journal might not). This needs to be defined.
- 2) What biological questions do you hope to answer by building a pan-genome? The authors give boilerplate text about virulence and AMR but never use the data to provide any insight into these major problems.
- 3) No mention of the vast body of prior art/knowledge on pan-genome software (e.g., Roary, PanOCT, OrthoMCL, and a dozen others) and approaches and why none of them will work for *Enterobacter*?
- 4) No mention anywhere of the published *Enterobacter* pan-genomes.
- 5) cd-hit is used to construct the pan-genome clusters, not the poorly described machine learning approach. Again, mSpectrum is for general readership in microbiology, not computer scientists.
- 6) Resolution of pan-genome structure is prevented by using cd-hit, which collapses (or groups together) paralogs. Paralogs are very important strategy for evolving new protein functions. The new functions provided by paralogs help to define new bacterial strains along with flexible genomic islands, which are also lost in this approach. Chromosomal rearrangements are also lost in this approach, which we know is also important in strain evolution.
- 7) Phylons are never described/defined. They don't represent strain resolution or species for that matter as there were genomes of 9 species used as input, but only 8 phylons (of 19) were species. The other phylons were groupings of mobile elements and other things. We don't know exactly because the list of genes was not provided as supplemental data.
- 8) MASH is a very fast and reliable tool for computing a proxy for ANI, but its reliability weakens with short distance between genomes. PMC6269478 found that fastANI was better than MASH at separating closely related species into strains (>99.9% ANI).
- 9) The set of genomes used does not at all represent the total number of genomes available from NCBI. BV-BRC is a great resource, but does not capture every genome like they say they do. As of Aug 2025, there are 1887 *Enterobacter* genomes with 1078 being *E. hormaechei*.

Minor critiques:

- 1) Line 15: What does the structure of the pan-genome tell us about *Enterobacter*?
- 2) Line 16: Because many of the genomes used are draft genomes (i.e., in pieces), they are not complete by any means. Using "complete" is misleading.
- 3) Lines 20-21: The data presented does not show that Phylons are consistent with MASH clusters as there was not a 1:1 look-up of MASH to Phylon or Phylon to known species-level assignments.
- 4) Lines 21-22: Again, there are more phylons than phylogroups, which is not consistent.
- 5) Line 24: Phylons represent some grouping of genes for sure, but they do not define the structure of the pan-genome in a biologically meaningful way it is unclear what they represent exactly (species for some, mobile DNA bin for others). Draft genomes were used, preventing the true structure of the pan-genome from being known (i.e., missing genes at contig breaks, rearrangements impossible to map), paralogs were collapsed preventing strain-level gene expansion from being counted in the structure.
- 6) Lines 25-29: The use of strains is questionable here as no definition of a strain was used here nor were any other available software tools used to validate strain assignments (e.g., strainEST, mixtureS, and others).
- 7) Line 32: define ESKAPEE acronym the first time used.
- 8) Line 35: remove the word "complete" because draft genomes were used. Also, the machine learning technique used here is not novel. It was used by the same group in a prior publication in mSphere on *E. coli* genomes.
- 9) Line 37: "deconvolute" should be "define", but I don't see anywhere in the results on all forms of horizontal transmission of genetic information used by *Enterobacter* (i.e., phages vs plasmids vs IS elements, flexible genomic islands). From what I read, the whole paper is a high level summary of binned genes.

- 10) Lines 69-70: similar studies on *Enterobacter* were not referenced.
- 11) Line 70: What key questions? How will "filling these gaps" enhance clinical practice? There was never any regard for comparing phylons to clinical outcomes.
- 12) In general, the Introduction is very fluffy and high-level with injustice to prior software, key limitations to pangenome analyses, and prior *Enterobacter* pan-genome studies. Pangenome should also be defined and some text describing the NMF machine learning approach that a general audience can understand. Perhaps decomposition is not the correct term to use. It seems more like a process of binning or classification rather than breaking something down. You are making a structure, not destroying it.
- 13) Throughout the manuscript, bacterial genus and species names must be italicized, even in the reference section!
- 14) Line 84: the BV-BRC was the only data source, so "public domain" is misleading as there are many more *Enterobacter* genomes (even more unassembled ones in the SRA that could have been used) available from NCBI, where BV-BRC pulls their data from.
- 15) Line 89: Please describe the distribution of plasmid replication, incompatibility and ability to conjugate (part of deconvoluting horizontal inheritance) with something like MobSuite.
- 15) Line 91: "significant proportion" How do you define significant here?
- 16) Line 111: MASH does not do well with strain-level resolution. It is not clear what MASH parameters were used, but you will need many more than 1000 k-mers to get enough data points to resolve strain-level distance (< 0.1).
- 17) Lines 113-114: It has already been noted that *E. cloacae* has been misclassified (see reference 12).
- 18) Line 516: similarity should be ">" 80% not "<"
- 19) Line 524: What were the MASH parameters used (# and size of kmers, ANI cut-off)?
- 20) Line 542: "all" should be "no"
- 21) Lines 607-610: Please provide the list of genes and annotation for each phylon

The manuscript “Structure of the Enterobacter Pan-Genome is Revealed Using Machine Learning” demonstrates that NMF-based accessory genome clustering, “Phylon”, provides a powerful, most likely reproducible, and biologically meaningful approach to classify *Enterobacter* species and subspecies, particularly for challenging groups such as *E. hormaechei*. It provides genome-scale, trait-based definitions that can reconcile or refine ambiguous or contentious taxonomic groupings. However, the manuscript lacks several key points:

1. The study focused on exploring the distribution of genetic traits across the species, but it ignores some of the important species. Recent work has formally described *E. pasteurii* and *E. quasihormaechei* using full genome sequencing and resolved their phylogenetic distinction from *E. hormaechei* and its subspecies. These lineages are now recognized as valid species with available reference genomes.
2. The exclusion of these species from the genus-wide pangenome analysis may be due to restricting genomes to older entries labeled under *E. hormaechei* or the ECC, without subsequent confirmation or update.
3. The introduction section lacks details on the taxonomic challenges within the genus *Enterobacter*. Since the genome data was apparently downloaded in October 2023, it is likely that some genomes are misnamed or misclassified.
4. Using *E. coli* pangenome studies as a basis for genus-wide pangenome analysis in *Enterobacter* does not seem appropriate.
5. Selecting only complete genomes can introduce bias into the study. The authors should be aware of this limitation and should discuss how it may affect their conclusions.
6. Why was only the BV-BRC database used for downloading the genome dataset? Since October 2023, the *Enterobacter cloacae* complex has undergone taxonomic revisions, and new species such as *E. quasihormaechei* and *E. pasteurii* have been described and validly published.
7. Did the authors confirm species-level assignments for all genomes used in the pangenome analysis? This typically requires Average Nucleotide Identity (ANI), digital DNA-DNA hybridization (dDDH), and core gene phylogeny to ensure accurate taxonomic assignments for comparative genomics. Initial identifications based on 16S or NCBI/BV-BRC labels can be inaccurate. Inclusion of misidentified genomes would weaken the pangenome conclusions, especially in complexes with ongoing taxonomic revisions.

Some minor comments:

- L577: What are these single-copy genes? Please provide a list. How do these genes differ from other universal core bacterial genes identified by UBCG and GTDB-Tk tools?
- L579: Why was a sequence distance tree constructed, rather than using a maximum likelihood (ML) tree method?

We thank the editor for the opportunity to revise and resubmit our manuscript and thank the reviewers for detailed and constructive comments. We have taken all these comments into account to revise the manuscript into a significantly improved version. Among other updates we have updated our strain collection (64% increase), improved species designation of our strains and updated our pangenome construction software by utilizing Panaroo to improve pangenome structure and identify paralogs. We also addressed all instances in the manuscript text that were unclear or needed clarifications. We respond to each point below inline in blue text. We have also included a revised version of the manuscript with changes tracked to ease re-review. We look forward to hearing comments on this revised version.

Summary of Major Changes:

Major improvements to this manuscript include an update to the dataset used for construction of the pangenome and downstream analysis. The original dataset consisted of 473 complete genomes; an additional 304 unique genome assemblies from the NCBI Genomes database have now been added, resulting in a total of 777 unique genome assemblies being included in the new dataset, increasing the size of the dataset by 64%. Reviewer discussion of the dataset raised concerns over usage of incomplete strains in the pangenome and the downstream NMF workflow, however both our original and updated dataset used for application of NMF include only complete genomes. Justification for this selection and the potential limitations of only using complete genomes have been included in the discussion section.

For pangenome construction, CD-HIT has been a robust component of pangenome construction workflows, however newer pangenome construction softwares offer additional features and capabilities which can improve the quality of a pangenome. Therefore, for this resubmission we utilized Panaroo for construction of the pangenome as Panaroo provides capabilities in gene refinding and paralog resolution. Additionally, rank selection methodology was made more robust through analysis of all ranks from one to 99. All of these factors combined to improve the quality of the dataset and pangenome used as input to our NMF machine learning method, and optimal rank selection for the NMF model. The number of Phylons computed using our method increased from 19 to 31 (63%), paralleling the increase in the size of our dataset. While the number of Phylons increased the underlying structure the Phylons represent is highly conserved even amongst the above changes and improvements. This conserved structure is visualized below with a Sankey diagram displaying the flow of strain affinities in the previous phylon structure to the affinities of these same strains to the new phylon structure. The biological insights derived from the Phylons in the originally submitted manuscript are preserved through the updated Phylons, including the subpopulations of *E. hormaechei subsp. stiegerwaltii* with additional motility genes, and computation of *mobile* Phylons which represent commonly acquired sets of genes through horizontal gene transfer, such as phylon *mobile-1* containing tellurite resistance genes. The conserved structure of the Phylons alongside the increased dataset size and improved pangenome construction methodology displays the robustness of our method for computation of Phylons, gene sets which represent major modes of inheritance across the accessory genome of *Enterobacter*.

In addition to these improvements, suggestions from reviewers leading to improvements in genome annotation and software selection have been incorporated. Software such as FastANI, FastTree, Genome Taxonomy Database Toolkit (GTDB-Tk), and the MOB Suite of plasmid annotation software have been incorporated into the paper, all of which have improved the interpretability of the results through improved annotation and curation of the pangenome, the details of which are described in the updated manuscript and in the responses to reviewer comments below. Overall, we believe that the updates incorporated into this manuscript have significantly improved the methodology and results of this manuscript. We would like to thank the editor and reviewers for the valuable comments and their consideration of this revised manuscript.

Affinity Flow from Previous to New NMF Structure

Affinity Flow from Previous NMF Structure to Updated NMF: Strains represented as flows from the old structure to the new are those which possessed affinity for a species-associated phylon in both the original NMF structure and the updated structure. All 15 phylons from the original structure are represented as an equivalent Phylon in the new structure or as multiple phylons representing additional resolution provided by the increased dataset and rank of the updated Phylon structure. Two of the Phylons in the updated structure (*mori* and *hormaechei-hoffmannii-3*) are representative of phylons which consist solely of strains which were not present in or did not possess phylon affinity in the previous Phylon structure.

Response to Reviewers

Editor's comment

Introduction

- Authors are invited to specify which are the elements of methodological novelty brought about by this work. For instance, they could reassess the methodological comparison with their own refs. 14-15 and appropriately modify the manuscript.

Thank you for this comment. Ref 14 (now 19) does not include any pan-genome decomposition, however ref 15 (now 20) is the first example of applying mathematical decomposition methods to the E. coli pangenome. In this manuscript we've extended the approach to a different organism (Enterobacter) and expanded the phylogenetic lens to the Genus scale from the Species scale. We also delved deeper into phylon error modes and characterized differences between phylons within Enterobacter, as well as providing methodology by which new genomes can be assessed computationally in the context of this pangenome and Phylon structure. These aspects make the study a significant expansion from ref 15 (now 20) that deserves publication in its own right.

- Lines 55-63. Which are the taxonomic advances repeatedly evoked by the authors? The authors are invited to detail their statements as they are relevant to capture the significance of the proposed work.

This section of the introduction section has been significantly revised in order to highlight the value of other pangenomic studies

- Lines 70-80 enumerate a sequence of potential future benefits coming from the application of pangenome analysis tools. However, the Introduction section be focused on the present manuscript content.

Thank you for this comment, other reviewers also noted we should include references to past work on Enterobacter genomes. Thus we have significantly updated the introduction. We've moved future benefits to the discussion section.

- Acronyms should be described at the first instance (ESKAPEE)

A definition of the acronym was added at the first usage in the manuscript (line 37).

Results

- Lines 90: Please rephrase "no species appeared to be overrepresented in plasmid possession"

The text was updated for improved clarity to read as follows: "no species possessed more plasmids than others across the genus."

- Figure 1 legend. Please, rephrase "detailed previously (see Methods)" as the Method section follows Figure 1. Please, correct the numbers reported in the legend as they are not consistent with the numbers displayed in Figure 1b. Figure 1c: Please, improve the visibility of strain colors. Figure 1d: "distinct" is an adjective whereas a verb is needed. Authors state "There are 9 species (with 5 subspecies of *E. hormaechei*) noted" but the figure does not show the subspecies. Figure 1e: Please, rephrase "Treemap of species distribution of dataset based on annotated species on BV-BRC." Authors are invited to remove Fig. 1e that is superfluous and not cited in the manuscript. Authors could place it in the Supplementary Material. Please, increase parallelly the readability of the heatmap in pane d).

Thank you for catching this. We have updated the figure content based on these comments as well as results from our updated dataset.

- How many clusters were obtained by CD-HIT?

In the original pangenome, among complete genomes there were 67,539 clusters and amongst all genomes there were 128,358 clusters.

Upon updating the pangenome to utilize Panaroo and increasing the number of complete genomes considered, there are a total of 58,421 gene clusters.

- If the core genome contains genes present in more than 97% of strains, this information is in contrast with Line 157 do the authors state that 430 genes are said to have one allele present in 50% or more of strains.

Gene clusters from CD-HIT consist of sequences with 80% or higher sequence identity. In this case we define an allele for a gene as an identical protein sequence. Therefore, each core gene may have multiple alleles. Among core genes, 430 of the core genes have one identical allele amongst all sequences in the cluster which is present in 50% or more of all strains in the pangenome. This statement is highlighting conservation of protein sequence of specific core genes beyond the determination of gene cluster membership by Panaroo.

- What is the functional annotation source of the pangenome?

Bakta is used for annotation and gene calling for all genomes, while EggNOG was used to annotate representative protein clusters from CD-HIT, now updated to Panaroo, please see the methods section for details on specific version and commands used.

- Do the authors consider only the representative genomes of MASH-based clusters in NMF? Please, clarify lines 205-207 and explain clearly which are the genome involved at the beginning of the section as it can improve readability.

No, we use all genomes for input to pangenome construction and NMF. We have updated these lines in the text to improve readability.

- Lines 219-224 let me think that phylons contain strains whereas they are made of genes. Please, clarify.

Sorry for the confusion on this point. Phylons are co-occurring gene sets. Strains carry phylons and are a mosaic linear combination of the phylons they carry (e.g. phylon1 + mge-phylon2 + mge-phylon3).

- How are false positives and false negatives computed in "Assessing the quality of the NMF decomposition"?

The original P matrix can be reconstructed by multiplication of the L and A binarized matrices. This leads to a binary matrix representing the reconstruction of P (termed P-prime or P_{rec}). The reconstructed P matrix is compared to the original P matrix. If a gene is determined to be absent in a strain in the reconstructed P but it is present in the original P, this is determined to be a false negative. If a gene is determined to be present in a strain in the reconstructed P but it is not present in the original P matrix, this is determined to be a false positive.

Methods - Consider Addressing these as manuscript re-writes occur

- N50 and L50 threshold. Please, provide references/justifications for the thresholds used.

We prioritized complete strains so chose genomes with an L50 score of 1 since complete strains should have 1 large circularized main chromosome that contains at least 50% of all the DNA. For N50 scores we took the reference strain's (*Enterobacter hormaechei* assembly ASM1904824v1) N50 score and required that all other strains have an N50 score at least 90% of that value. Additionally, Supplemental Figure 11 panel A was added for visualization of the distribution of N50 values for genomes filtered for downstream analysis.

- CheckM contamination <2.7% or Completeness <97.4%. Please, provide justifications.

We evaluated the CheckM contaminations and completeness metrics and picked the elbow of the plots to select a threshold cutoff. Description of this methodology was added to the methods

section of the paper: “Genomes with CheckM contamination <2.7% or Completeness <97.4% were removed. These thresholds for completeness and contamination were determined by determining the elbow and knee of the dataset, respectively, in order to filter outliers for contamination.”

- Please, provide reference for Patrick. How was it used?

Patric (now BV-BRC) is the NIH-NIAID bioinformatic resource center. It contains a collection of assembled genomes for key bacterial and viral species. We used this resource to identify genomes of *Enterobacter* for further consideration in this study. Here is a citation:

Introducing the Bacterial and Viral Bioinformatics Resource Center (BV-BRC): a resource combining PATRIC, IRD and ViPR.

Olson RD, Assaf R, Brettin T, Conrad N, Cucinell C, Davis JJ, Dempsey DM, Dickerman A, Dietrich EM, Kenyon RW, Kuscuoglu M, Lefkowitz EJ, Lu J, Machi D, Macken C, Mao C, Niewiadomska A, Nguyen M, Olsen GJ, Overbeek JC, Parrello B, Parrello V, Porter JS, Pusch GD, Shukla M, Singh I, Stewart L, Tan G, Thomas C, VanOeffelen M, Vonstein V, Wallace ZS, Warren AS, Wattam AR, Xia F, Yoo H, Zhang Y, Zmasek CM, Scheuermann RH, Stevens RL. *Nucleic Acids Res.* 2022 Nov 9:gkac1003. doi: 10.1093/nar/gkac1003. Epub ahead of print. PMID: 36350631

- Please, provide justifications for the allowed range of GC content

Justification for the allowed range of GC content was included in the manuscript. Additionally, Supplemental Figure 11 provides visualization of the distribution of GC content of genomes in panel B. The updated text is as follows: “GC content was limited to between 54 and 57%, as most *Enterobacter* species possess GC content between 54.5-56% and the above limits were selected for removal of outliers (Supplemental Fig. 11b).”

- Please, rephrase "as well as accessions with more than 20 annotated plasmids being removed as outliers"

Thank you for pointing this out. This sentence was rephrased in the text: “Accessions with more than 20 annotated plasmids being removed as outliers relative to the number of plasmids found in other genomes in the dataset”

- Authors must specify which genomic features provided by Bakta (coding, non-coding and so on so forth) are used in the subsequent analysis.

The pangenome is constructed using coding regions as annotated by Bakta, as well as regions refound using Panaroo’s gene refinding algorithm based on genomic context of genes. The methods text describing the use of Panaroo now also specifies this is a coding sequence pangenome: “The pangenome of coding sequences was constructed using Panaroo v1.5.2. The

core genome threshold was initially determined using CD-HIT to construct an initial pangenome structure.”

- Line 511-512. Authors are invited to rephrase the sentence and to specify the alignment percentage minimum cutoffs since citations are not sufficient.

These sentences were rephrased for improved clarity as follows: “For CD-HIT, cutoffs of 80% amino acid sequence similarity and 80% sequence alignment coverage were used. , in line with previous studies ¹⁵. Panaroo was run using the options ‘--clean-mode sensitive’, a family threshold of .8 in line with the CD-HIT protocol, and the core threshold set based on the above approximation from CD-HIT. The ‘--remove-invalid-genes’ option was used in order to enable the Bakta annotated files to be used as input for Panaroo.”

- How do the authors select the representative sequences in CD-HIT?

Selection of representative sequences for CD-HIT was determined by the longest sequence in a given cluster being selected as the representative sequence. A similar approach was chosen for selection of the representative sequence for Panaroo results: “Representative sequences from the pangenome were selected as the longest sequence without premature stop codons from each gene cluster computed by panaroo.”

- The BGC annotation is not clear. What is the pangenome which the authors refer to here? Please, clarify the insufficient description provided in lines 519-522.

Thank you for this comment. The BGC annotation from BGCFlow provides information about the presence of a BGC in a given strain, as well as which genes are determined as being part of that BGC. The second sentence, “Genes associated with a BGC in a given strain were then mapped to genes in the pangenome to determine which genes were associated with a given BGC family. ” This sentence attempted to describe the technical methodology in code of processing the outputs of the BGCFlow pipeline, however these details did not add necessary information to the methods section and only served to complicate understanding of the process of BGC annotation undertaken in this work. This sentence was thus removed for clarity.

- Lines 527. What does "from BV-BRC annotation." mean?

This sentence was modified for additional clarity: “Sequences with Mash distance greater than .205 being removed as being in the top 1% of greatest distance from the mean of representative sequences for *Enterobacter* species present in the pangenome (representative sequences based on representative for the *Enterobacter* genus listed on BV-BRC).”

- Lines 528-520. Sensitivity analysis was performed to optimize the threshold for clustering, resulting in a value of .15. Authors are invited to provide evidence (plots, tables) of this analysis.

Supplemental Figure 12 was added in order to display the selection of the threshold selected for determining the number of clusters. This served to be a conservative estimate of the elbow of the plot for number of clusters against the selected threshold for number of clusters determined against distance threshold.

- Is 0.15 a Pearson's correlation coefficient (PCC=)?

Yes, this is a Pearson's correlation coefficient (PCC). The threshold presented for the Mash clustering is in units of 1 - PCC, which is the distance value used for clustering.

- Why do the authors transform a MASH distance into a PCC and then the PCC into a distance?

This methodology of clustering Mash distances by first transforming them into a PCC and the a PCC distance has been utilized before in previous studies and has shown better clustering results as opposed to clustering the Mash distances directly:

Abram, K., Udaondo, Z., Bleker, C., Wanchai, V., Wassenaar, T. M., Robeson, M. S., & Ussery, D. W. (2021). Mash-based analyses of *Escherichia coli* genomes reveal 14 distinct phylogroups. *Communications Biology*, 4(1), 117. <https://doi.org/10.1038/s42003-020-01626-5>

Rajput, A., Chauhan, S. M., Mohite, O. S., Hyun, J. C., Ardalani, O., Jahn, L. J., Sommer, M. Oa., & Palsson, B. O. (2023). Pangenome analysis reveals the genetic basis for taxonomic classification of the *Lactobacillaceae* family. *Food Microbiology*, 115, 104334. <https://doi.org/10.1016/j.fm.2023.104334>

- Authors must clarify which are the strains used to annotate the MASH-based clusters? Lines 531-535 are unclear. What do the authors mean by "type strain"?

Based on suggestions from reviewers we have updated the assignment of taxonomic information to Mash clusters and is now performed using GTDB annotated species and subspecies information. The text is updated to reflect this change.

The original type strains were provided in the code repositories and are based on strain information provided by Table 1 of Reference 15, for reference. Here is the original table of strains for Mash cluster annotation:

species	type_strain	Strain Selection	
asburiae	1646339	https://f1000research.com/articles/7-521	
bugandensis	881260.3	https://f1000research.com/articles/7-521	
cancerogenou	69218.16	https://f1000research.com/articles/7-521	

s		
cloacae clade K	550.42	https://f1000research.com/articles/7-521
asburiae clade L	61645.63	https://f1000research.com/articles/7-521
cloacae clade N	550.1227	https://f1000research.com/articles/7-521
cloacae clade s	550.979	https://f1000research.com/articles/7-521
cancerogenous clade t	69218.15	https://f1000research.com/articles/7-521
cloacae cloacae	716541.4	https://f1000research.com/articles/7-521
cloacae dissolvens	1104326	https://f1000research.com/articles/7-521
hormaechei hoffmannii	1812934	https://f1000research.com/articles/7-521
hormaechei hormaechei	888063.8	https://f1000research.com/articles/7-521
hormaechei oharae	301102.4	https://f1000research.com/articles/7-521
hormaechei sterigerwaltii	299766.4	https://f1000research.com/articles/7-521
hormaechei xiangfangensis	1296536	https://f1000research.com/articles/7-521
kobei	208224.1	https://f1000research.com/articles/7-521
ludwigii	299767.2	https://f1000research.com/articles/7-521
mori	539813.4	BV-BRC Listed Reference Genome
rogenkampii	1812936	https://f1000research.com/articles/7-521

- Authors are invited to correct the definition of rare genes at lines 541-543

The sentence was updated in order to more accurately reflect the methodology used to define the rare genome: “From the inflection point of this curve, 90% of the distance from the inflection point to presence in no strains was used to determine the boundary of the rare genome, resulting in genes with <7% (58 strains) presence being placed into the rare genome.”

- At lines 545-561 the choice of the rank range and the choice of the final rank in non-negative matrix factorization are poorly described. What are the accuracy metrics mentioned? How are they used jointly with AIC? Authors are invited to explain the reason underlying binarization. The intermittent references to the previous article of the authors turned out inappropriate. Even though NMF is widely used, it is useful to introduce the formula linking L and A .

We apologize for describing these methods poorly. In this resubmission, the methodology used for rank selection has been significantly updated for more robust rank selection. We now sweep over all ranks from one to 99. Supplemental Figure 6 was updated to display metric performance for Akaike information criteria (AIC), Bayesian information criteria (BIC), residual sum of squares (RSS), and the Jaccard Heuristic (binarized) for AIC (JAIC-h) across all ranks. Additionally, the methods were updated with this information: "Rank was determined by running NMF across ranks one through 99 and computing accuracy metrics of each, including Bayesian information criterion, residual sum of squares, and Akaike information criterion (AIC). The final rank of 31 was selected as it was determined to be the knee, or point of diminishing returns, for increasing the rank in lowering the AIC."

The non-binarized data for the NMF results is included with the manuscript's data, however the reasoning behind binarization is to increase interpretability of the results by defining genes which have sufficient weighting in a Phylon as to constitute membership of a gene in a given Phylon.

Reviewer #2 (Comments for the Author):

The manuscript "Structure of the Enterobacter Pan-Genome is Revealed Using Machine Learning" demonstrates that NMF-based accessory genome clustering, "Phylon", provides a powerful, most likely reproducible, and biologically meaningful approach to classify Enterobacter species and subspecies, particularly for challenging groups such as *E. hormaechei*. It provides genome-scale, trait-based definitions that can reconcile or refine ambiguous or contentious taxonomic groupings. However, the manuscript lacks several key points, please the attachment.

1. The study focused on exploring the distribution of genetic traits across the species, but it ignores some of the important species. Recent work has formally described *E. pasteurii* and *E. quasihormaechei* using full genome sequencing and resolved their phylogenetic distinction from *E. hormaechei* and its subspecies. These lineages are now recognized as valid species with available reference genomes.

This is a valid concern and we have taken several steps to update the dataset with representatives from these recently described species and improve or description of the taxonomy of *Enterobacter* based on latest research. Additional complete genomes, collected as of the end of August 2025, have been added to the decomposition for the pangenome, resulting in the inclusion of several new species of *Enterobacter* into the pangenome structure. However, calculation of the Phylons using NMF relies on a sufficient number of strains of a given species to be present in the pangenome for Phylons to be extracted representing them. These valid species are not discussed in this study due to a lack of available data for their genetics in the population of *Enterobacter* genomes. As more representative genomes become available updates to the pangenome and NMF decomposition will become possible.

2. The exclusion of these species from the genus-wide pangenome analysis may be due to restricting genomes to older entries labeled under *E. hormaechei* or the ECC, without subsequent confirmation or update.

Confirmation of taxonomic information of strains included in the pangenome was performed using the Genome Taxonomy Database Toolkit (GTDB-Tk) for assignment of species-level classification of genomes included in the pangenome. Updating the complete strain dataset used in the pangenome prior to reconstruction of the pangenome also served to address issues with updates to both taxonomy and the available genomes of *Enterobacter*. Thank you for these suggestions.

3. The introduction section lacks details on the taxonomic challenges within the genus Enterobacter. Since the genome data was apparently downloaded in October 2023, it is likely that some genomes are misnamed or misclassified.

As described in the response to points 1 and 2, additional complete strains were added in order to remedy potential issues with taxonomic changes which have taken place since October 2023. Furthermore, we now use GTDB-Tk in order to examine the taxonomic labeling of

considered genomes. Additional references and details regarding the taxonomic changes occurring within the genus are described in the introduction section as well.

4. Using *E. coli* pangenome studies as a basis for genus-wide pangenome analysis in *Enterobacter* does not seem appropriate.

While E. coli and Enterobacter are very different organisms, lessons learned from pangenome construction and interrogation in E. coli pangenomic studies have established useful techniques and tools for analysis of other pangenomes from other microbes. This builds on previous work through application of useful pangenomic tools and machine learning methodologies for application and interpretation to Enterobacter in order to show their potential for useful adaptation to analysis at the genus scale.

5. Selecting only complete genomes can introduce bias into the study. The authors should be aware of this limitation and should discuss how it may affect their conclusions.

Complete genomes were selected in order to ensure maximal quality of the genomes and gene presence and absence used as input for the NMF decomposition; we acknowledge, however, that this does introduce potential sampling bias to the study as compared to all available genomes. We believe that this decision is justified in this study, but we also acknowledge the importance of recognizing this trade-off in the paper and have added lines 405-408 in the discussion in order to ensure this decision and its potential consequences are made evident to readers.

6. Why was only the BV-BRC database used for downloading the genome dataset? Since October 2023, the *Enterobacter cloacae* complex has undergone taxonomic revisions, and new species such as *E. quasihormaechei* and *E. pasteurii* have been described and validly published.

As described in the responses to 1-3, additional complete genomes were downloaded from NCBI and all unique genomes as of August 2025 were added to the pangenome in response to these reviews. This resulted in a 64% increase in the size of the dataset, as well as a resulting 63% increase in the number of phylons computed, which was also affected by improvements to the robustness of rank selection methods for NMF used in this manuscript. Some newly described species, as described in response to point 1, do not possess sufficient available genomes to result in Phylons in the final NMF decomposition and are therefore not discussed in this paper.

7. Did the authors confirm species-level assignments for all genomes used in the pangenome analysis? This typically requires Average Nucleotide Identity (ANI), digital DNA-DNA hybridization (dDDH), and core gene phylogeny to ensure accurate taxonomic assignments for comparative genomics. Initial identifications based on 16S or NCBI/BV-BRC labels can be inaccurate. Inclusion of misidentified genomes would weaken the pangenome conclusions, especially in complexes with ongoing taxonomic revisions.

This important point, as described above, was addressed by including the use of GTDB-Tk for annotation and classification of genomes as members of the *Enterobacter* genus in order to ensure robust conclusions can be reached in this study in line with up-to-date taxonomic information. Additionally, the use of Mash as a proxy for ANI (of which a Mash distance of .05 is an appropriate approximation of ANI of 95%) serves to compare within genomes selected for pangenomic analysis in order to ensure included genomes are sufficiently genetically similar as to be used for comparative genomic analysis of *Enterobacter*. Thank you for your suggestion of the inclusion of data from databases such as GTDB, as, while no genomes were filtered out through annotation using GTDB-Tk, it supported the filtration process undertaken using upstream QC and Mash filtration, as well as providing more robust taxonomic information for genomes used in this pangenomics study.

Citation for Mash:

Ondov, B. D., Treangen, T. J., Melsted, P., Mallonee, A. B., Bergman, N. H., Koren, S., & Phillippy, A. M. (2016). Mash: Fast genome and metagenome distance estimation using MinHash. *Genome Biology*, 17(1), 132. <https://doi.org/10.1186/s13059-016-0997-x>

Some minor comments:

- L577: What are these single-copy genes? Please provide a list. How do these genes differ from other universal core bacterial genes identified by UBCG and GTDB-Tk tools?

The use of single-copy core genes as originally described was updated in this manuscript in order to instead utilize a core-gene phylogeny (as aligned using MAFFT through Panaroo's functionality). This methodological switch in order to better reflect established methods for phylogenetic tree construction.

- L579: Why was a sequence distance tree constructed, rather than using a maximum likelihood (ML) tree method?

In addition to utilizing all core-genes for phylogenetic tree construction, FastTree, a maximum likelihood approximation method, was used for construction of the tree presented in Figure 4. The previously used distance methodology produced comparable results, however FastTree was selected for use in order to better reflect established methods for phylogenetic tree construction.

Reviewer #3 (Comments for the Author):

This manuscript provides an important analysis of the pan-genome of Enterobacter. Particularly, the authors use machine learning (non-negative matrix factorization) to identify clusters of accessory genes that best represent sub-groups of Enterobacter.

The paper is clearly written. I do appreciate the effort to provide biological interpretations for the different groups of accessory genes, especially on how they relate to AMR, virulence, and biochemical pathways.

Source data and methodologies are well-described and publicly available via Github.

Altogether, this will be an important resource for microbiologists and evolutionary biologists.

Very minor comments:

*What is the justification for the 7-97% cutoff on the definition of accessory genes?

The 7-97% cutoff for the accessory genome was determined using a curve-fitting method on the cumulative distribution of gene presence across the pangenome. This methodology utilizes the fitting of an S-shaped curve in order to determine the inflection points at which a subset of genes are present in nearly all strains (the core genome) and those which are present in very few strains (the rare genome). This methodology is described in the methods section of this paper and was established as a robust method for determining the boundaries of the core-accessory and rare genome in previous work.

Citation:

Hyun, J. C., Monk, J. M. & Palsson, B. O. Comparative pangenomics: analysis of 12 microbial pathogen pangenomes reveals conserved global structures of genetic and functional diversity. *BMC Genomics* 23, 7 (2022).

*Is the result of NMF decomposition unique? How robust is this to the cut-off definition above?

NMF as an algorithm does not give a unique decomposition necessarily, as the resultant matrices from the decomposition $V = WH$ can be scaled or permuted, resulting in “family” of solutions that are arbitrary scalings and permutations of one another. This is not necessarily a weakness of the decomposition, however, and it relates to our normalization of the resulting matrices for NMF works to account for potential scaling differences.

Referring to the robustness of the decomposition to the selection of cutoffs for the accessory genome, performed a sensitivity analysis comparing the normalized matrices of a variety of cutoffs with the final normalized matrices used in this study. All NMF decompositions were computed at rank 31, the final selected rank for this study, in order to compare directly between accessory genome thresholds. The cosine similarity of components is computed for the affinities

of strains to Phylons in the normalized matrix (as the input of strains is consistent while the gene input is different between different core and rare thresholds). A linear sum assignment methodology was used to select the best matching components (highest value unique matches of cosine similarity) and an average score was computed. This resulted in the following matrix of comparisons:

As can be seen in this matrix, the cosine similarity between all of these **L** and **A** matrices as compared to the final ones used in the study, those being 7.4-97.3%. Values of cosine similarity range from .80 to .97, all of which indicate a high level of similarity between the resultant matrices at all of these thresholds and the final Phylons computed for this study. In this, our method demonstrates the robust ability of NMF to find meaningful structure in the accessory genome in a way which is robust to selection of the boundaries of the accessory genome.

Citation regarding study performed on the uniqueness of NMF:

Laurberg, H., Christensen, M. G., Plumley, M. D., Hansen, L. K., & Jensen, S. H. (2008). Theorems on positive data: on the uniqueness of NMF. *Computational intelligence and neuroscience*, 2008, 764206. <https://doi.org/10.1155/2008/764206>

Reviewer #4 (Public repository details (Required)):

The authors must provide the list of genes that constitute each "phylon." Without proving this information, we are just taking their word that this approach any useful biological meaning over the vast plethora of prior pan-genome knowledge.

Thank you for bringing this important issue to our attention. While this information was able to be extracted from the data provided in the GitHub repositories, it was not immediately evident how that would be performed and it was certainly not easily accessible to those interested in exploring the Phylons who lacked computer science expertise. In order to address this issue, both of the GitHub repositories associated with this manuscript now include the list of genes in each Phylon. This information includes annotation information from EggNOG and if the gene is exclusive to a given phylon or not (that being that a given gene is present in only one specific Phylon). We hope that this increased availability of data will assist in its ability to be utilized by readers.

Reviewer #4 (Comments for the Author):

This paper describes the pan-genome of *Enterobacter* spp. using a machine learning approach; however, there are several major flaws with this paper that prevent it from being acceptable in mSpectrum.

1) What is a pan-genome (I know what it is, but many microbiologists that read this journal might not). This needs to be defined.

We appreciate the assistance in clarifying the importance of improving clarity of the introduction section for the audience of mSpectrum. Definition of the term pan-genome was added to the introduction before discussion of prior use of pangenomes for advancements in *Enterobacter* research.

2) What biological questions do you hope to answer by building a pan-genome? The authors give boilerplate text about virulence and AMR but never use the data to provide any insight into these major problems.

This pangenome provides information about the distribution of genes across a species that may be responsible for important traits in individual strains of *Enterobacter*, including metabolism, AMR and virulence. It also provides a gene-first mathematical structure of the pangenome, displaying the importance of the accessory genome in defining the differences between groups of strains. In this gene-first approach, it also reveals major subpopulations within species and subspecies, such as multiple Phylons for *E. hormaechei* *subsp. stiegerwaltii*, which reveal differences in accessory gene content across even closely related members of a given species or subspecies. Classifying components of the pangenome into core, accessory and rare components allows one to explore the vital nature of the rare genome on the acquisition of virulence and AMR genes. In contrast, the core and accessory genome possess a limited

number of unique AMR genes, while the rare genome contains the majority of all unique genes related to these traits of AMR and virulence. This manuscript and the pangenome and Phylon structure presented herein seek to define a mathematical structure (calculated using NMF) of the accessory genome which reflects a given genome as a summation of different lineage-associated Phylons and horizontally acquired genes (the mobile Phylons).

3) No mention of the vast body of prior art/knowledge on pan-genome software (e.g., Roary, PanOCT, OrthoMCL, and a dozen others) and approaches and why none of them will work for *Enterobacter*?

There may be confusion regarding our method. Constructing the pangenome itself is different from methods used to analyze the pangenome downstream. We originally used CD-HIT to construct our *Enterobacter* pangenome (references 14 and 15 in the original publication). However, as the reviewers pointed out, new pangenome construction software has recently been released with additional capabilities. Thus, in this submission we updated pangenome construction to use Panaroo, a pan-genome construction software which combines use of CD-HIT with gene-refinding and paralog separation capabilities. While multiple methods for pangenome construction have been shown to produce robust results, Panaroo was selected as a replacement for the previously utilized method for its wide acceptance and increased capabilities in handling paralogs. It is important to note, however, that our NMF approach is only applied after construction of the pangenome. It is used to analyze the results of pangenome construction software like Panaroo by mathematically decomposing the structure of the pangenome and thus is distinct from all of the pangenome software listed by the reviewer above. Our Phylon approach is a distinct and novel method to analyze the structure of a pangenome after it has been constructed using these other tools.

Panaroo Citation:

Tonkin-Hill, G., MacAlasdair, N., Ruis, C., Weimann, A., Horesh, G., Lees, J. A., Gladstone, R. A., Lo, S., Beaudoin, C., Floto, R. A., Frost, S. D. W., Corander, J., Bentley, S. D., & Parkhill, J. (2020). Producing polished prokaryotic pangenomes with the Panaroo pipeline. *Genome Biology*, 21(1), 180. <https://doi.org/10.1186/s13059-020-02090-4>

4) No mention anywhere of the published *Enterobacter* pan-genomes.

This is an important point and we apologize for neglecting to include references to past efforts. Additional text describing pangenomes in general was added to the introduction and references to published *Enterobacter* pangenomes were added in order to highlight the importance of this methodology for studying *Enterobacter* and the vital work which has already been done on that front. Therefore, this paragraph in the introduction was adjusted to include more information defining the term pangenome as well as adding additional references to existing efforts in *Enterobacter* pangenomics.

Additional references added:

Wu, W., Feng, Y. & Zong, Z. Precise Species Identification for Enterobacter: a Genome Sequence-Based Study with Reporting of Two Novel Species, *Enterobacter quasiroggenkampii* sp. nov. and *Enterobacter quasimori* sp. nov. *mSystems* 5, e00527-20 (2020).

Rahi, P., Mühle, E., Scandola, C., Touak, G. & Clermont, D. Genome sequence-based identification of *Enterobacter* strains and description of *Enterobacter pasteurii* sp. nov. *Microbiol. Spectr.* 12, e03150-23 (2024).

Kim, Y., Gu, C., Kim, H. U. & Lee, S. Y. Current status of pan-genome analysis for pathogenic bacteria. *Curr. Opin. Biotechnol.* 63, 54–62 (2020).

De Maayer, P., Green, T., Jordan, S., Smits, T. H. M. & Coutinho, T. A. Pan-genome analysis of the *Enterobacter hormaechei* complex highlights its genomic flexibility and pertinence as a multidrug resistant pathogen. *BMC Genomics* 26, 408 (2025).

Oni, F. I. et al. Whole-genome sequencing reveals *Enterobacter hormaechei* as a key bloodstream pathogen in six tertiary care hospitals in southwestern Nigeria. *Microb. Genomics* 11, (2025).

5) cd-hit is used to construct the pan-genome clusters, not the poorly described machine learning approach. Again, mSpectrum is for general readership in microbiology, not computer scientists.

Thank you again for highlighting the importance of writing the manuscript in a way that is clear to the readership of mSpectrum. Additional text describing the use of NMF and defining the term pangenome was added to the introduction in order to make the nature of the research more clear to the reader. Panaroo is now being used over a CD-HIT pangenome, and text describing the use of Panaroo for pangenome construction was added to the methods section in order to inform the reader of the specific method of constructing the pangenome matrix which was used for downstream machine learning (NMF) analysis. We have improved the description of this approach to better describe it for general readers.

6) Resolution of pan-genome structure is prevented by using cd-hit, which collapses (or groups together) paralogs. Paralogs are very important strategy for evolving new protein functions. The new functions provided by paralogs help to define new bacterial strains along with flexible genomic islands, which are also lost in this approach. Chromosomal rearrangements are also lost in this approach, which we know is also important in strain evolution.

This is an important point which we recognize as a significant consideration for this pangenomic analysis. As was mentioned in response to Reviewer #4's major concern #3, the pangenome has now been updated to use Panaroo instead of the previous CD-HIT method. Panaroo separates paralogs by using a genome-context method by building a pangenome graph of adjacent genes. This provides a genome-architecture aware framework for pangenome construction which helps to resolve the important issue of paralog detection and separation.

7) Phylons are never described/defined. They don't represent strain resolution or species for that matter as there were genomes of 9 species used as input, but only 8 phylons (of 19) were species. The other phylons were groupings of mobile elements and other things. We don't know exactly because the list of genes was not provided as supplemental data.

An additional sentence has been added to the results section upon the introduction of the term Phylons: "Phylons represent sets of genes which are co-occurring across strains, as defined by the **L** matrix with the **A** matrix representing a given genome's association with each Phylon." The reviewer is correct that the phylons do not represent a 1:1 analysis of species in the pangenome. They are instead an unsupervised discovery of major modes of genetic variation and inheritance in the accessory genome which define inheritance across the sequenced genomes of *Enterobacter*.

8) MASH is a very fast and reliable tool for computing a proxy for ANI, but its reliability weakens with short distance between genomes. PMC6269478 found that fastANI was better than MASH at separating closely related species into strains (>99.9% ANI).

FastANI was tested for comparison to Mash in order to ensure relative performance of Mash as compared to FastANI. The output of FastANI was compared to Mash by conversion to a distance metric (1 - ANI value) and then calculation of the cosine similarity between vectors was computed. The average cosine similarity between the Mash distances of one strain to all others and the equivalent vector of FastANI distances was calculated to be .997 (and a Pearson correlation of .978). This comparison indicates that selection of Mash does not provide meaningful difference between values computed using FastANI. FastANI data is, however, now included in the Zenodo repository linked in the data availability section of this manuscript for analysis. MASH was only used to classify species relationships for QC/QA filtering. We've now supplemented this with both GTDB and FastANI results.

9) The set of genomes used does not at all represent the total number of genomes available from NCBI. BV-BRC is a great resource, but does not capture every genome like they say they do. As of Aug 2025, there are 1887 *Enterobacter* genomes with 1078 being *E. hormaechei*.

For use in the calculation of the Phylons using NMF, complete genomes were specifically selected due to their higher reconstruction quality as compared to draft or fragmented genomes. However, to ensure that we have the latest set of complete genomes we did update and refresh our dataset for this resubmission. On NCBI, as of August 2025, there were a total of 982 unique genome assemblies for complete genomes of *Enterobacter*. This did represent a large increase in available data as compared to the original dataset presented. Accordingly, the additional genomes present on NCBI were processed for potential inclusion into the pangenome. Following quality control and filtration, the total size of the dataset was increased from 473 to 777, an increase of 64% to the size of the dataset which is now presented in the revised manuscript. Additionally, the number of phylons (both due to increased dataset size and more robust determination of the optimal NMF rank), increased from 19 to 31. This increase in the

size of the dataset now gives a more up-to-date representation of the available complete genomes for *Enterobacter*.

Minor critiques:

1) Line 15: What does the structure of the pan-genome tell us about Enterobacter?

We added to this sentence to highlight the value which can be gained from pangenomics research for a general audience : “The growing availability of publicly accessible *Enterobacter* genomes offers an opportunity to reveal the structure of its pangenome, uncovering the catalogue of genes across the genus and their distribution across the different species and subspecies of the genus.”

2) Line 16: Because many of the genomes used are draft genomes (i.e., in pieces), they are not complete by any means. Using "complete" is misleading.

We specifically selected for complete genomes that are not in pieces, see criteria for L50 scores in methods. For use in the calculation of the Phylons using NMF, complete genomes were specifically selected due to their higher reconstruction quality as compared to draft or fragmented genomes. The fragmented genomes discussed in the study were used after NMF decomposition. They were mapped onto the NMF structure in order to provide information about the application of the Phylon structure onto a broader set of *Enterobacter* genomes. In this, complete genomes were specifically used for this study.

3) Lines 20-21: The data presented does not show that Phylons are consistent with MASH clusters as there was not a 1:1 look-up of MASH to Phylon or Phylon to known species-level assignments.

We thank you for this comment and have attempted to better clarify what Phylons represent in this resubmission. Phylons represent mathematically determined (by NMF) modes of inheritance in the accessory genome, representing the major modes of inheritance, both lineage-associated and horizontal, across the accessory genome. We updated line 20 to read: “The Phylons are representative of major modes of inheritance, both lineage-associated and horizontal, found across the pangenome. Using NMF, we defined 31 Phylons, representative of 21 lineage-associated gene sets, and 10 Phylons containing genes associated with mobile genetic elements.” While there is not a 1:1 look-up of Phylon membership to MASH or species-level assignment (see response below), there is logical consistency in the Phylon structure that corresponds with taxonomic structure.

4) Lines 21-22: Again, there are more phylons than phylogroups, which is not consistent.

A useful result of NMF decomposition is identification of phylons associated both with lineages and with mobile elements. All strains are a linear combination of these phylons. Thus, yes it is possible to have more phylons than phylogroups (to represent these mobile phylons).

5) Line 24: Phylons represent some grouping of genes for sure, but they do not define the structure of the pan-genome in a biologically meaningful way it is unclear what they represent exactly (species for some, mobile DNA bin for others). Draft genomes were used, preventing the true structure of the pan-genome from being known (i.e., missing genes at contig breaks, rearrangements impossible to map), paralogs were collapsed preventing strain-level gene expansion from being counted in the structure.

Complete, closed genomes were selected specifically to avoid several of these concerns. Furthermore, issues regarding the use of complete genomes over draft genomes and using Panaroo to assist in resolving issues regarding paralogs were discussed in response to Reviewer #4's major concerns #3 and #9. The Phylons represent a mathematically determined structure of the presence and absence of genes in the pangenome using NMF, which is the structure this sentence is referring to.

6) Lines 25-29: The use of strains is questionable here as no definition of a strain was used here nor were any other available software tools used to validate strain assignments (e.g., strainEST, mixtureS, and others).

To avoid confusion we have updated the text to read "genomes" rather than strains. "NMF thus enabled phylogenetic and functional classification of strains based on the pangenome-scale assessment of a genome's gene portfolio."

7) Line 32: define ESKAPEE acronym the first time used.

Thank you for bringing this to our attention, a definition of the acronym was added at the first usage in the manuscript.

8) Line 35: remove the word "complete" because draft genomes were used. Also, the machine learning technique used here is not novel. It was used by the same group in a prior publication in mSphere on E. coli genomes.

The genomes selected for use in the NMF machine learning framework were complete genomes and additional draft genomes were then analyzed in the context of the structure established using the complete genomes.

The word "Novel" was removed from the Importance section as, while this work presents advancements in selection of NMF rank, advancements in software for method reuse and interpretation by users and reader, and application of the Phylon structure to incomplete genomes, this is indeed not the first application of NMF to pangenomics data. It is, however, the first application to Enterobacter genomes and a genus-level analysis.

9) Line 37: "deconvolute" should be "define", but I don't see anywhere in the results on all forms of horizontal transmission of genetic information used by *Enterobacter* (i.e., phages vs plasmids vs IS elements, flexible genomic islands). From what I read, the whole paper is a high level summary of binned genes.

The word "define" was substituted for "deconvolute" in order to clarify this sentence. Additional details regarding plasmid typing from MOB-suite was also added in order to enhance this element of the manuscript.

10) Lines 69-70: similar studies on *Enterobacter* were not referenced.

This sentence was removed and additional references to prior studies in *Enterobacter* pangenomics are now referenced in the preceding paragraph.

11) Line 70: What key questions? How will "filling these gaps" enhance clinical practice? There was never any regard for comparing phylons to clinical outcomes.

This sentence was re-written to better highlight the contributions of this manuscript to understanding of *Enterobacter* genetics: "Accurate delineation of core and accessory genes across the genus and the roles of these genes in determination of the traits of *Enterobacter* strains can enhance understanding of the defining genetic structure of the genus."

12) In general, the Introduction is very fluffy and high-level with injustice to prior software, key limitations to pangenome analyses, and prior *Enterobacter* pan-genome studies. Pangenome should also be defined and some text describing the NMF machine learning approach that a general audience can understand. Perhaps decomposition is not the correct term to use. It seems more like a process of binning or classification rather than breaking something down. You are making a structure, not destroying it.

Discussed in response to Reviewer #4's major concerns 1 and 4, definition for the term pangenome was added and reference to previous works of *Enterobacter* pangenomics were added.

Additional text in the introduction section was added to describe the application of NMF to the dataset: "Matrix factorization techniques enable the calculation of an interpretable low-dimensional structure from complex biological datasets [citation]. Non-negative matrix factorization (NMF) is a matrix factorization algorithm which extracts non-negative components which can additively combine to represent positive input data [citation]. Application of this technique can enable the genetic structure underlying various subpopulations of *Enterobacter* to be computed and analyzed." The term decomposition is used as an equivalent term to "factorization" as in non-negative matrix factorization. The term decomposition is indented to display the

13) Throughout the manuscript, bacterial genus and species names must be italicized, even in the reference section!

Thank you for bringing this important point of formatting to our attention. We went through the paper to ensure correct italicization was used throughout.

14) Line 84: the BV-BRC was the only data source, so "public domain" is misleading as there are many more Enterobacter genomes (even more unassembled ones in the SRA that could have been used) available from NCBI, where BV-BRC pulls their data from.

Also described in the response to Reviewer #4's major concern #9, the NCBI genomes database is also now being used in dataset acquisition. The methodology used for genome acquisition and selection are also described in the methods section and in this we believe that the term "public domain" is not misleading to a reader.

15) Line 89: Please describe the distribution of plasmid replication, incompatibility and ability to conjugate (part of deconvoluting horizontal inheritance) with something like MobSuite.

We thank you for this suggestion of software enhancements to include in the manuscript. MOB-suite was used to annotate and type the plasmids in the dataset and these details were added to the manuscript in order to enhance understanding of the distribution of plasmids in the dataset. The usage of this tool was also added to the methods section.

15) Line 91: "significant proportion" How do you define significant here?

The word "significant" in this context was not intended to imply a specific statistical meaning, therefore the wording was changed to "large proportion" and a percentage (24%) was given in order to make evident the number of samples which can be attributed to environmental isolation.

16) Line 111: MASH does not do well with strain-level resolution. It is not clear what MASH parameters were used, but you will need many more than 1000 k-mers to get enough data points to resolve strain-level distance (< 0.1).

Comparison of Mash to FastANI results were presented in the response to Reviewer #4's major concern #8. While Mash was used in the analysis for this paper, the results for FastANI are now also provided for analysis or use. Furthermore we were not concerned with strain-level resolution here but instead to ensure that genomes were of the Enterobacter genus.

17) Lines 113-114: It has already been noted that *E. cloacae* has been misclassified (see reference 12).

We agree that this text was redundant and it was accordingly removed. Use of the Genome Taxonomy Database (GTDB) and the associated toolkit (GTDB-Tk) was also included for

genome annotation and more accurate and up-to-date species assignment for strains included in the pangenome.

18) Line 516: similarity should be ">" 80% not "<"

Thank you for pointing this out, this issue was fixed in the text.

19) Line 524: What were the MASH parameters used (# and size of kmers, ANI cut-off)?

The parameters for the usage of Mash were added into the methods section of the manuscript: "The Mash distance calculation command was run using kmers of size 21 and a sketch size of 1,000."

20) Line 542: "all" should be "no"

Thank you for bringing this to our attention, the text was updated accordingly.

21) Lines 607-610: Please provide the list of genes and annotation for each phylon

As discussed in the response to major concerns regarding the manuscript, Phylon genes and annotation information are now included in the repositories associated with this manuscript.

Re: Spectrum01922-25R1 (Structure of the Enterobacter Pan-Genome is Revealed Using Machine Learning)

Dear Prof. Bernhard O Palsson:

Your manuscript has been accepted, and I am forwarding it to the ASM production staff for publication. Your paper will first be checked to make sure all elements meet the technical requirements. ASM staff will contact you if anything needs to be revised before copyediting and production can begin. Otherwise, you will be notified when your proofs are ready to be viewed.

Sincerely,
Angela Re
Editor
Microbiology Spectrum

Reviewer #2 (Comments for the Author):

Overall there is an improvement with the updated database and taxonomy. However, I still believe that the phylon assignments without considering the representative of all current valid sopecies might lead to errors. Particularly, in the E. hormaechei group, where authors defined 12 Phylons across 5 subspecies. It is likely that some of the genes shared by the strains in the current phylons might be also present in the some of the strains of closely replated species i.e E. pasteurii and E. quasihormaechei. In the present manuscript, there is no information to avoide such situations. Atleast, efforst should have been made to include one of the genomes (may be type strains) to avoide anomalies occuered due to the low taxon representation.

Reviewer #3 (Comments for the Author):

The authors have already addressed my comments from the previous round.